# Logical Expressiveness of Graph Neural Networks with Hierarchical Node Individualization

**Arie Soeteman**[1]    **Balder ten Cate**[1]
[1]Institute for Logic, Language and Computation
University of Amsterdam

## Abstract

We propose and study *Hierarchical Ego Graph Neural Networks* (HE-GNNs), an expressive extension of graph neural networks (GNNs) with hierarchical node individualization, inspired by the Individualization-Refinement paradigm for isomorphism testing. HE-GNNs generalize subgraph-GNNs and form a hierarchy of increasingly expressive models that, in the limit, distinguish graphs up to isomorphism. We show that, over graphs of bounded degree, the expressive power of HE-GNN node classifiers equals that of graded hybrid logic. This characterization enables us to relate the separating power of HE-GNNs to that of higher-order GNNs, GNNs enriched with local homomorphism count features, and color refinement algorithms based on Individualization-Refinement. Our experimental results confirm the practical feasibility of HE-GNNs and show benefits in comparison with traditional GNN architectures, both with and without local homomorphism count features.

## 1   Introduction

Graph neural networks (GNNs), and specifically message-passing neural networks [22, 25], are a dominant approach for representation learning on graph-structured data [49, 55]. Since the expressive power of GNNs within the message-passing framework is limited by the one-dimensional Weisfeiler-Leman test (WL) [38, 50], numerous architectures have been proposed with increased separating power—the ability to distinguish pairs of nodes or graphs. The separating power of deterministic isomorphism invariant GNN architectures can be studied logically. Morris et al. [38] and Barcelo et al. [6], building on earlier results by Cai et al. [9], gave logical characterizations of the separating power of standard message-passing GNNs, which matches Graded Modal Logic. Higher-order $k$-GNNs [38] have the separating power of $(k-1)$-WL [38], which equals that of First-Order Logic with $k$ variables and counting quantifiers. This yields a hierarchy of increasingly expressive models that separate nodes in graphs of size $n$ up to isomorphism when $k \geq n$. The logical study of GNNs has addressed fundamental questions about their expressive power [1, 23, 26, 46], convergence behavior [2], decidability [7], and explainability [48].

In this work, we introduce *Hierarchical Ego* GNNs (HE-GNNs), a fully isomorphism-invariant graph learning model inspired by the individualization-refinement (IR) paradigm used by graph isomorphism solvers, where node individualization is alternated with simple message-passing. We study the separating power of HE-GNN in detail through a logical lens, and make explicit connections with IR and existing expressive GNNs. Our main contributions are:

- We characterize the separating power of HE-GNNs with and without subgraph restrictions. Specifically, we provide logical characterizations in graded hybrid logic, situate the separating power of HE-GNNs within the WL hierarchy and show that HE-GNNs constitute a strict hierarchy in terms of their nesting depth $d$. HE-GNNs are able to separate nodes in graphs of size $n$ up to isomorphism when $d \geq n$, like higher-order GNNs, but at lower cost in terms of space complexity.

39th Conference on Neural Information Processing Systems (NeurIPS 2025).

- We prove that the graph separating power of HE-GNNs with depth $d$ is lower bounded by IR with $d$ rounds of invidualization, and show that common subgraph-GNNs are special cases of depth-1 HE-GNNs, so that our results characterize the separating power of subgraph models. We further identify a class of graphs with "small ego-rank" from which low-depth HE-GNNs can compute homomorphism counts. These results generalize and shed new light on known relations between subgraph-GNN, higher-order GNNs and homomorphism count vectors from tree-like graphs.

- We confirm empirically that HE-GNNs up to depth 2 improve performance on ZINC-12k and can distinguish strongly regular graphs beyond the reach of 3-GNNs, common subgraph GNNs and probabilistic IR methods.

Our results add foundational insights for understanding GNNs from a logical perspective, guiding the development of models with improved expressiveness.

**Related work**    Numerous deterministic message-passing architectures have been developed that implement isomorphism-invariant graph learning with increased separating power. Notable examples are higher-order networks [33, 37, 38], subgraph-GNNs [13, 51, 53] and GNNs augmented with homomorphism counts [5, 29]. These models allow for extensive analyses in terms of separating power and comparison with isomorphism invariant classifiers such as logics and the WL hierarchy (see [36]), and we compare them in detail with HE-GNNs in section 5.

In another line of work, node embeddings representing structural information are added before message-passing, often with probabilistic methods. Examples are positional encodings from Laplacian eigenvectors [16, 18, 27, 31], random sampling [1, 45] and random walks [43]. GNNs with randomly sampled features have complete separating power but are only isomorphism invariant in expectation: isomorphic graphs do not yield the same output but only the same distribution. Relevant to the current work are node embeddings inspired by IR, such as random traversals of IR trees [19] and Tinhofer orderings [24, 40]. These methods use non-determinism sparingly, yielding isomorphism-invariance for WL distinguishable graphs and compact graphs respectively. Another graph learning model in the IR paradigm was developed by Dupty and Lee [15], which includes several strong approximations that improve efficiency at the expense of invariance.

## 2    Background and notation

For a set $X$, we denote by $\mathcal{M}(X)$ the collection of all finite multisets of elements of $X$. We write $\oplus$ for vector concatenation, and $\delta_{xy}$ for Kronecker's delta (i.e., $\delta_{xy}$ is 1 if $x = y$ and 0 otherwise).

**Graphs**    Fix a set $P$ of binary node features. By a *graph* we will mean a triple $G = (V, E, \text{lab})$ where $V$ is set of *nodes*, $E \subseteq V \times V$ is a symmetric and irreflexive *edge* relation, and $\text{lab} : V \to 2^P$. The *degree* of a node $v$ in a graph $G = (V, E, \text{lab})$ is $|\{u \mid (v, u) \in E\}|$ and the degree of a graph is the maximum degree of its nodes. A *pointed graph* is a pair $(G, v)$ where $G$ is a graph and $v$ is a node of $G$. An *isomorphism* between graphs $G = (V, E, \text{lab})$ and $G' = (V', E', \text{lab}')$ is a bijection $f : V \to V'$ such that $E' = \{(f(v), f(u)) \mid (v, u) \in E\}$ and such that, for all $v \in V$, $\text{lab}'(f(v)) = \text{lab}(v)$. We write $G \cong G'$ if such a bijection exists. An isomorphism between pointed graphs $(G, v)$ and $(G', v')$ is defined similarly, with the additional requirement that $f(v) = v'$. By a *node classifier* we will mean a function $\text{cls}$ from pointed graphs to $\{0, 1\}$ that is isomorphism invariant (i.e., such that $\text{cls}(G, v) = \text{cls}(G', v')$ whenever $(G, v)$ and $(G', v')$ are isomorphic).

**Graph neural networks**    Let $D, D' \in \mathbb{N}$. A *graph neural network with input dimension $D$ and output dimension $D'$* (henceforth: $(D, D')$-GNN) is a tuple $((\text{COM}_i)_{i=1,\dots L}, (\text{AGG}_i)_{i=1,\dots L})$ with $L \geq 1$, where, for $1 < i \leq L$, $\text{COM}_i : \mathbb{R}^{2D_i} \to \mathbb{R}^{D_{i+1}}$, $\text{AGG}_i : \mathcal{M}(\mathbb{R}^{D_i}) \to \mathbb{R}^{D_i}$ with $D_1 = D$, $D_i \geq 1$, and $D_{L+1} = D'$. Each such GNN induces a mapping from embeddings to embeddings. More precisely, by a $D$-dimensional *embedding* for a graph $G = (V, E, \text{lab})$ we will mean a function $\text{emb} : V \to \mathbb{R}^D$. A $(D, D')$-GNN $\mathcal{A}$ defines a function $\text{run}_{\mathcal{A}}$ from a graph with a $D$-dimensional embedding to a $D'$-dimensional embedding as follows:

```
1: function run_𝒜(G, emb)
2:     emb⁰ := emb
3:     for i = 1, . . . , L do
4:         embⁱ := {v : COMᵢ(embⁱ⁻¹(v) ⊕ AGGᵢ({{embⁱ⁻¹(u) | (v, u) ∈ E}})) | v ∈ V}
5:     return emb^L
```

Every $(D, D')$-GNN $\mathcal{A}$ with $D = |P|$ and $D' = 1$ naturally gives rise to a node classifier $\mathsf{cls}_\mathcal{A}$. To define it, for a graph $G = (V, E, \mathrm{lab})$, let its *multi-hot label encoding* be the $|P|$-dimensional embedding $\mathrm{emb}_G$ given by $\mathrm{emb}_G(v) = \langle r_1, \ldots, r_n \rangle$ with $r_i = 1$ if $p_i \in \mathrm{lab}(v)$ and $r_i = 0$ otherwise. Then $\mathsf{cls}_\mathcal{A}(G, v) = 1$ if $\mathrm{run}_\mathcal{A}(G, \mathrm{emb}_G)(v) > 0$, and $\mathsf{cls}_\mathcal{A}(G, v) = 0$ otherwise. We denote the set of all such node classifiers $\mathsf{cls}_\mathcal{A}$ also by GNN.

The above definition of GNNs does not specify how the functions $\mathrm{COM}_i$ and $\mathrm{AGG}_i$ are specified. In practice, there are a few common choices for $\mathrm{AGG}_i$, namely *Sum*, *Min*, *Max* and *Mean*. These have the expected, point-wise, definition. For instance, *Sum* maps multisets of $\mathbb{R}^D$-tuples to $\mathbb{R}^D$-tuples by summing point-wise. As for the functions $\mathrm{COM}_i$, these are commonly implemented by a fully-connected feed-forward neural network (FFNN) using an activation function such as *ReLU*. Some of our results apply to GNNs with specific aggregation and combination functions, in which case this will be explicitly indicated. Otherwise, the results apply to all GNNs.

**Graded modal logic** The formulas of graded modal logic (GML) are given by the recursive grammar $\phi ::= p \mid \top \mid \phi \wedge \psi \mid \neg \phi \mid \Diamond^{\geq k} \phi$, where $p \in P$ and $k \in \mathbb{N}$. *Satisfaction* of such a formula at a node $v$ in a graph $G$ (denoted: $G, v \models \phi$) is defined inductively, as usual, where $G, v \models p$ iff $p \in \mathrm{lab}(v)$, the Boolean operators have the standard interpretation, and $G, v \models \Diamond^{\geq k} \phi$ iff $|\{u \mid (v, u) \in E \text{ and } G, u \models \phi\}| \geq k$. We use $\Diamond \phi$ as shorthand for $\Diamond^{\geq 1} \phi$ and we use $\Box \phi$ as a shorthand for $\neg \Diamond \neg \phi$. Every GML-formula $\phi$ gives rise to a node classifier $\mathsf{cls}_\phi$ where $\mathsf{cls}_\phi(G, v) = 1$ if $G, v \models \phi$ and $\mathsf{cls}_\phi(G, v) = 0$ otherwise.

**Example 2.1.** *Consider the* GML*-formula* $\phi = \Diamond^{\geq 2} \top \wedge \Box p$. *Then* $\mathsf{cls}_\phi(G, v) = 1$ *precisely if the node $v$ has at least two successors and all its successors are labeled $p$.*

**Weisfeiler Leman** Fix a countably infinite set of colors $\mathcal{C}$. A *node coloring* for a graph $G = (V, E, \mathrm{lab})$ is a map $\mathrm{col} : V \to \mathcal{C}$. A coloring is *discrete* if for all $v$ in $V$ $\mathrm{col}^{-1}(\mathrm{col}(v)) = v$. By a *colored graph* we mean a graph together with a coloring. The Weisfeiler Leman (WL) algorithm takes as input a colored graph $(G, \mathrm{col})$ and an integer $d \geq 0$. It produces a new coloring for the same graph as follows, where HASH is a perfect hash function onto the space of colors:

```
1: function WL(G, col, d)
2:     col⁰ = col
3:     for i = 1, . . . , d do
4:         colⁱ := {v : HASH(colⁱ⁻¹(v), {{colⁱ⁻¹(u)|(v, u) ∈ E}}) | v ∈ V}
5:     return col^d
```

For a graph $G = (V, E, \mathrm{lab})$, by the *initial coloring* of $G$ we will mean the coloring $\mathrm{col}_G$ given by $\mathrm{col}_G(v) = \mathrm{HASH}(\mathrm{lab}(v))$. We write $\mathrm{WL}(G, d)$ as a shorthand for $\mathrm{WL}(G, \mathrm{col}_G, d)$. In other words, $\mathrm{WL}(G, d)$ denotes the coloring obtained by starting with the initial coloring and applying $d$ iterations of the algorithm. Two pointed graphs $(G, v)$, $(G', v')$ are said to be *WL-indistinguishable* (denoted also $(G, v) \equiv_{\mathrm{WL}} (G', v')$) if $v$ and $v'$ receive the same color after $d$ iterations for $d = \max\{|G|, |G'|\}$ —that is, if $\mathrm{WL}(G, d)(v) = \mathrm{WL}(G', d)(v')$.

The WL algorithm gives rise to a node classifier for each $d \geq 0$ and subset $S \subseteq \mathcal{C}$, where $\mathsf{cls}_{d,S}^{\mathrm{WL}}(G, v) = 1$ if $\mathrm{WL}(G, d)(v) \in S$ and $\mathsf{cls}_{d,S}^{\mathrm{WL}}(G, v) = 0$ otherwise. Note that, by definition, such classifiers cannot distinguish WL-indistinguishable pointed graphs.

**Three-way equivalence** Given a collection $C$ of node classifiers (e.g., all GNN-based node classifiers), we denote by $\rho(C)$ the equivalence where $((G, v), (G', v')) \in \rho(C)$ if and only if, for all $\mathsf{cls} \in C$, $\mathsf{cls}(G, v) = \mathsf{cls}(G', v')$. In other words, $\rho(C)$ captures the expressive power of $C$ as measured by the ability to distinguish different inputs.

**Theorem 2.2.** $\rho(\mathrm{GNN}) = \rho(\mathrm{GML}) = \rho(\mathrm{WL})$

The equivalence in separating power between GNNs and WL was proven independently by Xu et al. [50] and Morris et al. [38]. Their equivalence with GML was shown by Barcelo et al. [6]. Indeed, it was shown in [6] that for every GML-formula, there is a GNN that implements the same node classifier:

**Proposition 2.3.** *([6]) For every* GML*-formula $\phi$ there is a GNN $\mathcal{A}$ such that* $\mathsf{cls}_{\mathcal{A}} = \mathsf{cls}_{\phi}$*. Moreover, the* GNN *in question only uses Sum as aggregation and a single ReLU-FFNN as combination function.*

The converse does not hold in general, but it does when we bound the degree of the input graph:

**Proposition 2.4.** *Let $\mathcal{A}$ be a $(D, D')$-GNN with $D = |P|$, let $N > 0$, and let*

$$X = \{\mathrm{run}_{\mathcal{A}}(G, \mathrm{emb}_G)(v) \mid G = (V, E, \mathrm{lab}) \text{ is a graph of degree at most } N \text{ and } v \in V\}.$$

*In other words $X \subseteq \mathbb{R}^{D'}$ is the set of all node embeddings that $\mathcal{A}$ can produce when run on a graph of degree at most $N$. Then $X$ is a finite set, and for each $\mathbf{x} \in X$, there is a GML-formula $\phi_{\mathbf{x}}$ such that for every pointed graph $(G, v)$ of degree at most $N$, it holds that $G, v \models \phi_{\mathbf{x}}$ iff $\mathrm{run}_{\mathcal{A}}(G, \mathrm{emb}_G)(v) = \mathbf{x}$. In particular, for each GNN-classifier $\mathsf{cls}_{\mathcal{A}}$ there is a GML-formula $\phi$ such that $\mathsf{cls}_{\phi}(G, v) = \mathsf{cls}_{\mathcal{A}}(G, v)$ for all pointed graphs $(G, v)$ of degree at most $N$.*

Proofs for these two propositions are provided in the appendix, as we will build on them.

## 3   Hierarchical Ego GNNs

In this section, we introduce and study the basic model of Hierarchical Ego GNNs (HE-GNNs). In the next section, we will further refine the model by means of subgraph restrictions.

**Hierarchical Ego GNNs**

- A $(D, D')$-HE-GNN of nesting depth $0$ is simply a $(D, D')$-GNN.
- A $(D, D')$-HE-GNN of nesting depth $d > 0$ is a pair $(\mathcal{B}, \mathcal{C})$ where $\mathcal{B}$ is a $(D + 1, D'')$-HE-GNN of nesting depth $d - 1$ and $\mathcal{C}$ is a $(D + D'', D')$-GNN.

A HE-GNN $\mathcal{A} = (\mathcal{B}, \mathcal{C})$ defines a function $\mathrm{run}_{\mathcal{A}}$ from embedded graphs to embeddings as follows:

---

1: **function** $\mathrm{run}_{\mathcal{A}}(G, \mathrm{emb})$
2:     **for** each node $v$ of $G$ **do**
3:         $\mathrm{emb}'(v) := \mathrm{emb}(v) \oplus \mathrm{run}_{\mathcal{B}}(G, \{u : \mathrm{emb}(u) \oplus \delta_{uv} \mid u \in V\})(v)$
4:     **return** $\mathrm{run}_{\mathcal{C}}(G, \mathrm{emb}')$

---

In other words, for each node $v$, we run $\mathcal{B}$ after extending the node embeddings to uniquely mark $v$, and concatenate the resulting embedding for $v$ to its original embedding. After constructing a new embedding for each $v$ we run $\mathcal{C}$. On a graph with $n$ nodes a HE-GNN with depth $d$ and $l$ layers at each depth generates $O(n^d)$ graphs with different unique labelings. Since an $l$ layer GNN sends at most $l \cdot n^2$ messages, a HE-GNN with depth $d$ and $l$ layers at each depth sends $O(l \cdot n^{d+2})$ messages. Applying the individualizations in depth-first order, $(d + 1) \cdot n$ node embeddings are stored, which is bounded by $n^2$. Naturally the time and space complexity depend on the embedding dimension and the chosen aggregation and combination functions. Just as in the case of GNNs, each $(D, D')$-HE-GNN $\mathcal{A}$ with $D = |P|$ and $D' = 1$ naturally gives rise to a node classifier $\mathsf{cls}_{\mathcal{A}}$.

**Example 3.1.** *Let $\mathcal{B}$ be a $3$ layer $(2, 2)$-GNN with element-wise sum as aggregation and the identity map as combination. Let $\mathcal{C}$ be a trivial $(2, 2)$-GNN that doesn't change the input. Then $\mathcal{A} = (\mathcal{B}, \mathcal{C})$ is a $(1, 2)$-HE-GNN of depth $1$ such that $\mathrm{run}_{\mathcal{A}}(G, v) \neq \mathrm{run}_{\mathcal{A}}(G', v')$ for the graphs in figure 1.*

This shows that HE-GNN with nesting depth $1$ has strictly more separating power than GNN. Let HE-GNN-$d$ denote all classifiers $\mathsf{cls}_{\mathcal{A}}$ where $\mathcal{A}$ is a HE-GNN of nesting depth $d$. As we will see below, HE-GNN-$d$ in fact forms an infinite hierarchy with respect to separating power for increasing values of $d$. To show this, we first give a logical characterization of HE-GNN-$d$.

**Graded hybrid logic**  Graded hybrid logic (henceforth GML($\downarrow$)) extends GML with *variables* and the *variable binder* $\downarrow$. To be precise, the formulas of GML($\downarrow$) are generated by the grammar $\phi ::= p \mid x \mid \neg\phi \mid \phi \wedge \psi \mid \Diamond^{\geq k}\phi \mid \downarrow x.\phi$. We restrict attention to *sentences*, i.e., formulas without free variables. The definition of satisfaction for a GML-formula at a node $v$ of a graph $G = (V, E, \mathrm{lab})$, extends naturally to GML($\downarrow$)-sentences as follows: $G, v \models \downarrow x.\phi$ if $G[x \mapsto v], v \models \phi$, where $G[x \mapsto v]$ denotes a copy of $G$ in which $x$ is treated as a binary node feature true only at $v$. By the $\downarrow$-*nesting-depth* of a GML($\downarrow$)-sentence, we will mean the maximal nesting of $\downarrow$ operators in the sentence. We denote with GML ($\downarrow^d$) all sentences with maximal $\downarrow$-nesting-depth $d$.

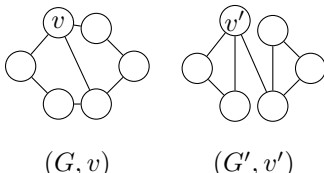

$(G, v) \qquad (G', v')$

Figure 1: Two non-isomorphic pointed graphs that are WL-indistinguishable.

**Example 3.2.** *The sentence $\phi = \downarrow x.\Diamond\Diamond\Diamond x$, which has $\downarrow$-nesting-depth 1, is satisfied by a pointed graph $(G, v)$ precisely if $v$ lies on a triangle. In particular, considering the example in Figure 1, $\phi$ distinguishes $(G, v)$ from $(G', v')$. This also shows that GML($\downarrow$) is more expressive than GML.*

**Example 3.3.** *Building on the above example, the sentence $\psi = \downarrow x.\Diamond(\phi \wedge \Diamond(\phi \wedge \Diamond\phi \wedge \Diamond(\phi \wedge x)))$, which has $\downarrow$-nesting-depth 2, is satisfied by $(G, v)$ precisely if $v$ lies (homomorphically) on a cycle of length 4 consisting of nodes that each lie on a triangle.*

In the literature, hybrid logics often include an @ operator, where $@_x\phi$ states that $\phi$ holds at the world denoted by the variable $x$. Over undirected graphs, however, every GML($\downarrow$, @)-sentence is already equivalent to a GML($\downarrow$)-sentence of the same $\downarrow$-nesting-depth.

The connection between GNNs and GML described in the previous section extends to a connection between HE-GNNs and GML($\downarrow$):

**Theorem 3.4.**  $\rho(\text{HE-GNN}) = \rho(\text{GML}(\downarrow))$. *Moreover, for $d \geq 0$, $\rho(\text{HE-GNN-}d) = \rho(\text{GML}(\downarrow^d))$.*

The proof, given in the appendix, is along the lines of Propositions 2.3 and 2.4. Indeed, there is a translation from GML($\downarrow$)-sentences to HE-GNNs, and, conversely, over bounded-degree inputs, there is a translation from HE-GNNs to GML($\downarrow$)-sentences. Both translations preserve nesting depth. In Section 5, we put this logical characterization to use to obtain a number of further results.

## 4   Hierarchical Ego GNNs with subgraph restriction

Since HE-GNNs generate an exponential number of graphs in $d$, we make the models more manageable by restricting subgraphs to a fixed radius $r$ around the uniquely marked node, in line with the common approach for subgraph-GNNs ([20, 51, 52, 53]).

**Hierarchical Ego Subgraph-GNNs**

- A $(D, D')$-HES-GNN of depth 0 is simply a $(D, D')$-GNN.
- A $(D, D')$-HES-GNN of depth $d > 0$ is a triple $(\mathcal{B}, \mathcal{C}, r)$ where $\mathcal{B}$ is a $(D + 1, D'')$-HES-GNN of depth $d - 1$, $\mathcal{C}$ is a $(D + D'', D')$-GNN, and $r$ is a positive integer.

Given a graph $G = (V, E, \mathrm{lab})$, a node $v \in V$, and a positive integer $r$, we will denote by $G_v^r$ the induced subgraph of $G$ containing the radius-$r$ neighborhood of $v$.

---

1: **function** run$_\mathcal{A}(G, \mathrm{emb})$          # *For $\mathcal{A} = (\mathcal{B}, \mathcal{C}, r)$*
2:     **for** each node $v$ of $G$ **do**
3:         $\mathrm{emb}'(v) := \mathrm{emb}(v) \oplus \mathrm{run}_\mathcal{B}(G_v^r, \{u : \mathrm{emb}(u) \oplus \delta_{uv} \mid u \in V\})(v)$
4:     **return** run$_\mathcal{C}(G, \mathrm{emb}')$

---

There is only one change compared to the previous version: $G$ got replaced by $G_v^r$ on line 3. Now if a HES-GNN with depth $d$, radius $r$ and at most $l$ layers is applied to a graph with degree $k$, it generates $O(n \cdot (k^r + 1)^{max(0,d-1)})$ graphs, sends at most $l \cdot n^2$ times as many messages and stores $n + d \cdot (k^r + 1)$ embeddings. Each $(D, D')$-HES-GNN $\mathcal{A}$ again gives rise to a node classifier $\mathcal{A}$. We denote with HES-GNN-$r$ the set of such classifiers for radius $r$ and with HES-GNN-$(d,r)$ the restriction to nesting depth $d$.

**Graded hybrid subgraph logic**  GML($\downarrow$, $W$) further extends GML($\downarrow$) with a *"within" operator* $W^r$ inspired by temporal logics with forgettable past [3, 10]. The formulas of GML($\downarrow$, $W$) are generated by $\phi ::= p \mid x \mid \neg\phi \mid \phi \land \psi \mid \Diamond^{\geq k}\phi \mid \downarrow x.\phi \mid W^r\phi$ . The definition of satisfaction for GML($\downarrow$)-sentences is extended by letting $G, v \models W^r\phi$ if $G_v^r, v \models \phi$. We will use $\downarrow_{W^r} x.\phi$ as a shorthand for $\downarrow x.W^r\phi$. We denote with GML ($\downarrow_W$) the fragment of GML ($\downarrow$, $W$) in which $\downarrow$ and $W$ can only be used in this specific combination with each other, and we denote with GML ($\downarrow_{W^r}^d$) (for specific integers $d$ and $r$), the further fragment with radius $r$ and where $\downarrow_{W^r}$ can be nested at most $d$ times. In terms of separating power, GML($\downarrow_W$) is equivalent to GML($\downarrow$), but pairing variable binders with subgraph restrictions of a specific radius serves to decrease expressive power. The connection between HE-GNN and GML($\downarrow$) we established in Theorem 3.4 now extends to the case with subgraph restrictions:

**Theorem 4.1.**

1. $\rho(\text{HES-GNN}) = \rho(\text{GML}(\downarrow_W)) = \rho(\text{GML}(\downarrow)) = \rho(\text{HE-GNN})$.

2. $\rho(\text{HES-GNN-}r) = \rho(\text{GML}(\downarrow_{W^r}))$. *Moreover,* $\rho(\text{HES-GNN-}(d,r)) = \rho(\text{GML}(\downarrow_{W^r}^d))$.

This is established again through a uniform translation from GML($\downarrow_{W^r}^d$) sentences to HES-GNN-$(d,r)$ classifiers and a converse uniform translation over bounded degree inputs from HES-GNN-$(d,r)$ classifiers to GML($\downarrow_{W^r}^d$) sentences.

The separating power of HES-GNN-$(d,r)$ classifiers strictly increases with $d$ and $r$. Note that $\rho(X) \subsetneq \rho(Y)$ means that $X$ has strictly more separating power than $Y$:

**Theorem 4.2.** *For $d \geq 1, r \geq 0$,* $\rho(\text{HES-GNN-}(d,r+1)) \subsetneq \rho(\text{HES-GNN-}(d,r))$.

**Theorem 4.3.** *For $d \geq 0$ and $r \geq 3$,* $\rho(\text{HES-GNN-}(d+1,r)) \subsetneq \rho(\text{HES-GNN-}(d,r))$

## 5 Comparison with other models

In this section, we build on the logical characterizations from the previous sections to obtain a number of technical results, drawing connections between isomorphism testing and several GNN architectures by comparing their expressive power with that of HE-GNNs and HES-GNNs.

### 5.1 Relationship with Individualization-Refinement

The individualization and refinement (IR) paradigm is applied by all state-of-the-art graph isomorphism solvers [14, 30, 34, 41]. In its usual presentation (e.g. [35]), *color refinement*, *cell-selection* and *individualization* procedures are applied in alternating fashion, resulting in a tree whose nodes are labeled by increasingly refined colorings of the input graph, with discrete colorings at the leafs. In practice, some further optimizations are typically implemented, exploiting symmetries to further reduce the size of the tree, but these optimizations do not affect the result of the equivalence test, and we do not consider them here. In order to make a precise comparison, we define WL-IR:

---

1: **function** WL-IR($G, \text{col}, d$)
2:   $\text{col}' := \text{WL}(G, \text{col}, |G|)$
3:   **if** $\text{col}'$ is discrete or $d = 0$ **then**
4:     **return** ($\text{col}'$)
5:   **else**
6:     Let $c$ be the least color with $|\text{col}'^{-1}(c)| \geq 2$, and let $\text{col}'^{-1}(c) = \{v_1, \ldots, v_n\}$,
      For each $i \leq n$, let $\text{col}'_i := \{v : \text{HASH}(\text{col}'(v), \delta_{vv_i}) \mid v \in V\}$

7:     **return** (  $\overset{\text{col}'}{\diagup \mid \diagdown}$  )
        WL-IR($G, \text{col}'_1, d-1$)  $\ldots$  WL-IR($G, \text{col}'_n, d-1$)

---

Here we use WL as the refinement procedure (which is indeed common practice) and apply a simple cell-selection procedure that assumes an order on the set of colors and picks the least non-singleton color. We include an extra input parameter $d$ that controls the number of individualization steps

the algorithm can perform. WL-IR is guaranteed to produce discrete colorings when choosing $d \geq |G|$. We write WL-IR$(G, d)$ as shorthand for WL-IR$(G, \mathrm{col}_G, d)$. We can now compare two graphs by choosing suitable $d$ and testing if WL-IR$(G, d)$ and WL-IR$(G', d)$ yield isomorphic trees. Let $G \equiv_{\text{WL-IR-}d} G'$ if and only if this comparison does not distinguish $G$ from $G'$. For $d = 0$, it is clear that $\equiv_{\text{WL-IR-}d}$ corresponds simply to indistinguishability by the Weisfeiler Leman test. If $d$ is sufficiently large, on the other hand, each leaf of the tree is labeled by a discrete coloring, so that WL-IR distinguishes graphs up to isomorphism:

**Proposition 5.1.** *Let $G, G'$ be graphs and $d \geq min(|G|, |G'|)$. Then $G \equiv_{\text{WL-IR-}d} G'$ iff $G \cong G'$.*

Thus, by varying $d$ we obtain a family of increasingly refined equivalence relations for graphs. In order to relate these equivalence relations to those induced by HE-GNNs of different nesting depths, we must first overcome a technical issue. WL-IR is designed to compare graphs, not nodes. Let $G \equiv_{\text{cls}} G'$ if $\{\!\{\mathsf{cls}(G, v) \mid v \in V\}\!\} = \{\!\{\mathsf{cls}(G', v) \mid v \in V'\}\!\}$, and $G \equiv_C G'$ if $G \equiv_{\text{cls}} G'$ for all classifiers $\mathsf{cls}$ in $C$. The graph separating power of WL-IR with depth 0 matches that of GNN.

**Proposition 5.2.** *$G \not\equiv_{\text{WL-IR-0}} G'$ if and only if $G \not\equiv_{\text{GNN}} G'$*

The graph separating power of WL-IR-$d$ is a lower bound to that of HE-GNN-$d$ for $d \geq 0$

**Theorem 5.3.** *For $d \geq 0$, if $G \not\equiv_{\text{WL-IR-}d} G'$, then $G \not\equiv_{\text{HE-GNN-}d} G'$*

In fact, for connected graphs, and with depth $d + 1$, HE-GNN node classifiers already suffice, since the separating power of local message-passing matches global message-passing over connected individualized graphs [52]:

**Theorem 5.4.** *Let $(G, v), (G', v')$ be connected pointed graphs and let $d \geq 0$. If $G \not\equiv_{\text{WL-IR-}d} G'$, there exists a depth $d + 1$ HE-GNN $\mathcal{A}$ such that $\mathsf{cls}_{\mathcal{A}}(G, v) \neq \mathsf{cls}_{\mathcal{A}}(G', v')$.*

Proposition 5.1 and theorems 5.3, 5.4 show that for sufficiently large $d$, HE-GNN-$d$ classifiers distinguish graphs up to isomorphism. Contrary to WL-IR, HE-GNNs combine individualized graphs hierarchically. We show this increases separating power when $d = 1$:

**Theorem 5.5.** *There exist $G, G'$ such that $G =_{\text{WL-IR-1}} G'$ but $G \not\equiv_{\text{HE-GNN-1}} G'$*

Using recent results by Rattan and Seppelt [44], theorem 5.5 implies that HE-GNN-1 is strictly more separating than cospectrality of adjacency, Laplacian and Seidel matrices. In addition, this shows that the hierarchical message-passing scheme of HE-GNN-1 adds to expressive power, compared to aggregating over all individualized graphs in parallel. It remains open if the same holds for all $d \geq 1$.

## 5.2 Relationship with homomorphism count enriched GNNs

In [5], the authors assume a finite set of rooted graphs $\mathfrak{F} = \{F_1, \dots F_k\}$ and, given an input graph $G$, for each node $v$, they add the finite homomorphism count vector $\hom(\mathfrak{F}, (G, v))$ to the initial embedding of $v$ before running a GNN. Here, by a *rooted graph* we mean a pointed graph $(F, u)$ that is connected, i.e., such that every node of $F$ is reachable from $u$. Note that the input dimensionality of the GNN is thus assumed to be $|P| + k$ instead of $|P|$. This increases the expressive power of the model. For example, the non-isomorphic nodes in Figure 1 can be distinguished from each other by including the cycle of length 3 (with a distinguished node) as a pointed graph in $\mathfrak{F}$. We will refer to a GNN that runs over a $\mathfrak{F}$ enriched graph simply as a $\mathfrak{F}$-GNN.

**Theorem 5.6.** *Let $\mathfrak{F}$ be any finite set of rooted graphs each with at most $d$ nodes. Then there is a $(|P|, |P| + |\mathfrak{F}|)$-HE-GNN $\mathcal{A}$ of nesting depth $d$ such that, for all pointed graphs $(G, v)$,*

$$\mathrm{run}_{\mathcal{A}}(G)(v) = \mathrm{emb}_G(v) \oplus \hom(\mathfrak{F}, (G, v))$$

*The HE-GNN in question only uses Sum as aggregation and ReLu-FFNNs as combination functions.*

In particular, this shows that every $\mathfrak{F}$-GNN is equivalent to a HE-GNN. [1] The practical value of the above result, however, is limited by the fact that it requires a high nesting depth. As it turns out, for many choices of $\mathfrak{F}$, a very small nesting depth suffices.

---

[1] Theorem 5.4 in [11] is somewhat related as it states that the node separating power of homomorphism counts from all rooted graph is captured by hybrid logic (with @-operator but without counting modalities).

We will call a rooted graph $(F, u)$ *c-acyclic* if every cycle of $F$ passes through $u$. C-acyclicity is a relaxation of acyclicity, and c-acyclic rooted graphs can be thought of as trees with back-edges. Our next result will imply that when $\mathfrak{F}$ consists of c-acyclic structures, a nesting depth of 1 suffices. In order to state it in full generality, we need to introduce some further terminology. In particular, we introduce the notion of *ego-rank*.

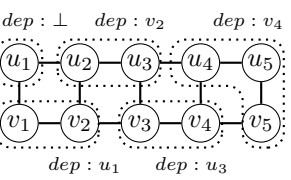

Figure 2: Rooted $5{\times}2$-grid (root: $u_1$)

Given a rooted graph $(G, v)$, let $dep : V \to V \cup \{\bot\}$ be a partial function from nodes to nodes and let $deps(u) = \{dep(u), dep(dep(u)), \ldots\} \setminus \{\bot\}$ be the (finite) set of nodes $u$ transitively "depends on". We require of the function $dep$ that:

1. $dep(v) = \bot$.

2. If $(w, u) \in E$, then $dep(w) = dep(u)$ or $w \in deps(u)$ or $u \in deps(w)$,

3. Every set of nodes with the same $dep$-value induces an acyclic subgraph.

The *ego-rank* of $(G, v)$ is the smallest value of the maximum node rank, where the node rank of a node $u$ is $|deps(u)|$, across all ways to choose the function $dep$ subject to the above constraints.

**Proposition 5.7.** *For all rooted graphs $(G, v)$ with $G = (V, E, \mathrm{lab})$,*

1. *tree-width$(G) - 1 \leq$ ego-rank$(G, v) \leq |V|$.*

2. *ego-rank$(G, v) = 0$ if and only if $G$ is acyclic.*

3. *ego-rank$(G, v) = 1$ whenever $(G, v)$ is c-acyclic.*

The ego-rank of a rooted graph is not upper-bounded by any function in its tree-width, and can be exponential in its tree-depth, as follows from the following example:

**Example 5.8.** *The rooted graph consisting of an $n \times 2$-grid, with one of its corners as the root, as depicted in Figure 2, has ego-rank $n - 1$ for $n \geq 1$.*

**Theorem 5.9.** *Let $\mathfrak{F}$ be any finite set of rooted graphs, let $d = \max\{ego\text{-}rank(F, u) \mid (F, u) \in \mathfrak{F}\}$.*

1. *Then there is a $(|P|, |P| + |\mathfrak{F}|)$-HE-GNN $\mathcal{A}$ of nesting depth $d$ such that, for all pointed graphs $(G, v)$, $\mathrm{run}_\mathcal{A}(G)(v) = \mathrm{emb}_G(v) \oplus \mathrm{hom}(\mathfrak{F}, (G, v))$. The HE-GNN uses multiplication in the combination functions.*

2. *For each $N > 0$, there is a $(|P|, |P| + |\mathfrak{F}|)$-HE-GNN $\mathcal{A}$ of nesting depth $d$ such that, for pointed graphs $(G, v)$ of degree at most $N$, $\mathrm{run}_\mathcal{A}(G)(v) = \mathrm{emb}_G(v) \oplus \mathrm{hom}(\mathfrak{F}, (G, v))$. The HE-GNN uses only Sum as aggregation and ReLu-FFNNs as combination functions.*

It follows that, in a non-uniform sense, every $\mathfrak{F}$-GNN is equivalent to a HE-GNN of nesting depth $\max\{ego\text{-}rank(F, u) \mid (F, u) \in \mathfrak{F}\}$ using only Sum and ReLu-FFNNs.

## 5.3  Relationship with higher-order GNNs

$k$-GNNs, as proposed by Morris et al. [38], apply message-passing between node subsets of size $k$, where subsets are adjacent when they share exactly $k - 1$ nodes. The separating power of these models is characterized by the $k$-variable fragment of first-order logic with counting quantifiers $\mathsf{C}^k$.

**Theorem 5.10** ([9, 38]). $\rho(k\text{-GNN}) = \rho(\mathsf{C}^k)$

$k$-GNNs become strictly more expressive with larger $k$, and distinguish graphs of size $n \leq k$ up to isomorphism. As such, they have proven to be a useful yardstick for the development of expressive GNN architectures. The model proposed by Morris et al. requires exponential space and time since it maintains $n^k$ features in memory, and applies message passing over $n^{k+1}$ edges. Maron et al. [33] proposed $k$-IGNs with $O(n^{k-1})$ space complexity and $O(n^k)$ time complexity, which were shown by Azizian and LeLarge [4] and Geerts [21] to have the same separating power as $k$-GNNs.

HE-GNNs constitute an alternative hierarchy, where nesting depth $d$ yields classifiers that distinguish graphs of size $n \leq d$ up to isomorphism (Theorem 5.3). HE-GNNs perform simple message passing on $n^d$ graphs, but they can do this in sequence and hence need to store only $n^2$ node features. Using the logical characterizations we show the separating power of HE-GNNs with nesting depth $d$ is at most that of $d + 2$-GNNs, or equivalently the $d + 1$-WL algorithm.

**Theorem 5.11.** *For $d \geq 0$, $\rho((d+1)\text{-WL}) = \rho((d+2)\text{-GNN}) \subseteq \rho(\text{HE-GNN-}d)$.*

This generalizes a recent result by Frasca et al. [20], who showed that the separating power of many common subgraph-GNNs is bounded by that of 2-WL. We further show that this result is optimal, in the sense that HE-GNNs with nesting depth $d$ can distinguish nodes that $d + 1$-GNNs can not:

**Theorem 5.12.** *For $d \geq 0$, HES-GNN-($d$,3) can distinguish pointed graphs that cannot be distinguished by $d$-WL, or equivalently, by a $(d+1)$-GNN.*

Given the difference in the number of stored embeddings, one would expect that for $d \geq 0$, $\rho((d+1)\text{-WL}) \subsetneq \rho(\text{HE-GNN-}d)$. We leave this as an open conjecture.

### 5.4 Relationship with other variants of subgraph-GNNs

Numerous recent studies [20, 51, 52, 53, 54] have proposed variants of subgraph-GNNs, where message passing is applied to a collection of subgraphs. Subgraph-GNNs show state-of-the-art performance on real-world benchmarks such as ZINC molecular property prediction [8, 20, 54]. In particular, a variant of subgraph-GNNs in which nodes receive neighborhood embeddings based on their radius $r$ neighborhood with distinguished center node label, also known as "ego-networks", have become prominent as simple yet expressive subgraph-GNN architecture [20, 51, 53].

ID-GNNs [51] are HES-GNNs $(\mathcal{B}, \mathcal{C})$ with nesting depth 1, where $\mathcal{C}$ is a trivial GNN that doesn't apply any message passing. Nested GNNs [53] do not use individualization, but perform global pooling over subgraphs followed by message passing over the input graph. Since local aggregation with individualization is strictly more expressive than global aggregation over connected subgraphs [52], the separating power of nested GNN is strictly less than that of HES-GNNs with nesting depth 1. Theorem 4.1 thus provides a logical upper bound to the separating power of these models.

Several other generalizations of subgraph-GNNs have recently been proposed [39, 42]. Most related to this work, Qian et al. introduced Ordered Subgraph Aggregation networks that apply message passing on $|G|^k$ copies of $G$, each labeled with the atomic type of a $k$ size subgraph, and perform aggregation over representations for a predefined selection of the $|G|^k$ subgraphs. Like HE-GNN this constitutes a strict hierarchy with separating power upper bounded by $k + 1$-WL, but not by $k$-WL. It is not immediately clear whether OSAN yields more expressive node classifiers than HE-GNN, since OSAN performs global aggregation over arbitrary sets of subgraphs, while HE-GNN uses local message-passing hierarchically.

## 6 Experiments

We apply HES-GNN to ZINC-12k [17, 28] and compare with standard GCN [32], GIN [50] and PNA [12] layers, as well as GINs augmented with random identifiers (GNN+RID) [1, 45], Tinhofer orderings (GNN+T) [40], and homomorphism counts from cycles of size 3 to 10 ($\mathfrak{F}$-GNN) [5]. We use feature dimension 256 for all models for a maximum of 1000 epochs on a single 20GB gpu. All code used for the experiments is available on git.[2] Table 1 shows the achieved mean absolute error after 10 runs and validation score selection. HES-GNNs with depth 1 outperform $\mathcal{F}$-GNN even though ZINC has cycle counts in its target function. HES-GNN-(2,3) performs equally well as $\mathcal{F}$-HES-GNN-(1,3), a depth 1 model augmented with homomorphisms counts.

Figure 3 shows the performance of HE-GNN and HES-GNN on distinguishing strongly regular graphs. We use 4 synthetic datasets, each containing 30 random isomorphisms of 10 strongly regular graphs for a specific choice of parameters $(v, k, \lambda, \mu)$ [47]. Since strongly regular graph are indistinguishable by 2-WL and hence by 3-GNNs, depth 1 HE-GNN and ID-GNN [51] perform at chance level. Adding cycle counts doesn't alleviate this, in line with theorem 5.9. Tinhofer orderings fully individualize all nodes to yield maximal separating power, resulting in 100% precision on the test set. However, since these orderings are not isomorphism-invariant over the tested graphs, T-GNN does not generalize to the test set. Depth 2 HE-GNNs distinguish the strongly regular graphs and generalize to unseen isomorphisms, even with restricted subgraph radius. Figure 4 shows time and memory usage of HE-GNNs with varying depth and radius on ZINC for a single forward pass.

---

[2]https://github.com/ariesoeteman/HEGNN

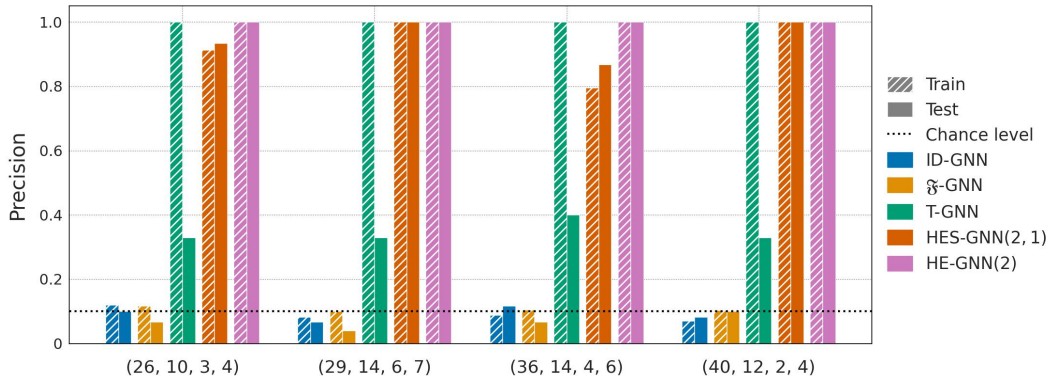

Figure 3: Precision on isomorphism classification of strongly regular graphs for four settings of $(v, k, \lambda, \mu)$. Models from left to right: ID-GNN [51], $\mathfrak{F}$-GNN with counts from $C_3, \dots C_{10}$ [5], GNN with Tinhofer labelings [40], HES-GNN with $d = 2, r = 1$ and HE-GNN with $d = 2$.

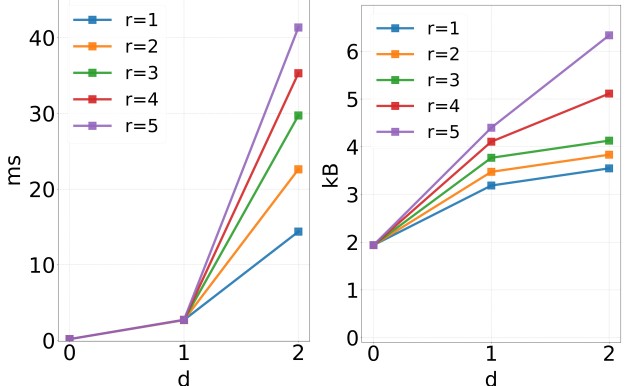

Figure 4: Time (left) and memory (right) per sample of a single forward pass of HES-GNN-$(d,r)$ on ZINC. This does not include gradients. Experiments run with batch size 20 and feature dimension 256.

Table 1: Mean absolute error on ZINC-12k, test scores after 10 runs.

| Model | | ZINC (MAE) |
|---|---|---|
| GCN [32] | | 0.216 |
| GIN [50] | | 0.193 |
| PNA [12] | | 0.207 |
| GNN+T [40] | | 0.199 |
| GNN+random ID [1, 45] | | 0.226 |
| $\mathfrak{F}$-GNN [5] | | 0.154 |
| | (d,r) | |
| HES-GNN | (1,1) | 0.191 |
| | (1,2) | 0.189 |
| | (1,3) | 0.149 |
| | (2,1) | 0.188 |
| | (2,2) | 0.183 |
| | (2,3) | **0.140** |
| $\mathfrak{F}$-HES-GNN | (1,3) | **0.140** |

## 7 Limitations and directions for further research

**Efficiency** Compared to $k$-GNNs, HE-GNNs store $|G|^2$ instead of $|G|^k$ node features. Nevertheless, they still require an exponential amount of message passing steps in the nesting depth $d$, rendering implementations infeasible for large $d$. IR-based graph isomorphism tests typically reduce tree size via informed cell selection and automorphism pruning, whereas Dupty and Lee [15] address scalability in a learning setting through compressed approximate trees. Further study is needed to explore how techniques from isomorphism testing can be adapted to graph representation learning, and how these optimizations affect expressive power.

**Expressive power** Some questions regarding separating power are left open by our results. In particular, we gave sufficient conditions for graphs from which HE-GNN-$d$ can count homomorphisms in terms of size and ego-ranks, but we have not shown these conditions are necessary. Our results are also limited in scope by the fact that they concern separation power and not the uniform expressibility of functions. Several uniform expressibility results have recently been obtained for GNNs (both relative to first-order logic [6], and in terms of Presburger logic [7]), and it remains to be seen if these results can be extended to HE-GNNs.

**Beyond expressive power** The empirical success of GNNs cannot be understood through expressive power alone, as it also depends on trainability aspects such as convergence and generalization. These remain to be explored more extensively, across GNN architectures and in the context of HE-GNNs.

## Acknowledgements

We thank Martin Grohe, Ronald de Haan, and Carsten Lutz for valuable discussions that contributed to the development of this work.

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

## Missing proofs

## A Proofs for section 2

**Proposition 2.3.** *([6]) For every* GML*-formula $\phi$ there is a* GNN $\mathcal{A}$ *such that* $\mathsf{cls}_{\mathcal{A}} = \mathsf{cls}_{\phi}$. *Moreover, the* GNN *in question only uses Sum as aggregation and a single ReLU-FFNN as combination function.*

*Proof.* Let $\Phi$ be the smallest set of GML-formulas containing $\phi$ that is closed under taking subformulas. The *operator depth* of a GML-formula is defined as follows: $\mathrm{depth}(p) = 0$, $\mathrm{depth}(\neg\phi) = \mathrm{depth}(\phi) + 1$, $\mathrm{depth}(\phi \wedge \psi) = \max\{\mathrm{depth}(\phi), \mathrm{depth}(\psi)\} + 1$ $\mathrm{depth}(\Diamond^{\geq n}\phi) = \mathrm{depth}(\phi) + 1$. Let $|\Phi| = k$ and let $L$ be the maximal operator depth of a formula in $\Phi$. Let

$$\mathcal{A} = ((\mathrm{COM}_i)_{i=1\ldots L}, (\mathrm{AGG}_i)_{i=1\ldots L})$$

where

- $\mathrm{COM}_1 : \mathbb{R}^{2|P|} \to \mathbb{R}^k$ given by $\mathrm{COM}_1(x_1, \ldots, x_{|P|}, y_1, \ldots, y_{|P|}) = (z_1, \ldots, z_k)$ with $z_i = x_j$ if $\phi_i$ is of the form $p_j$ and $z_i = 0$ otherwise.

- For $i > 1$, $\mathrm{COM}_i : \mathbb{R}^{2k} \to \mathbb{R}^k$ is given by $\mathrm{COM}_i(x_1, \ldots, x_k, y_1, \ldots y_k) = (z_1, \ldots, z_k)$ with

$$z_i = \begin{cases} x_i & \text{if } \phi_i \text{ is of the form } p \\ 1 - x_j & \text{if } \phi_i \text{ is of the form } \neg\phi_j \\ \max\{x_j, x_m\} & \text{if } \phi_i \text{ is of the form } \phi_j \wedge \phi_m \\ 1 \text{ if } y_j \geq n, 0 \text{ otherwise} & \text{if } \phi_i \text{ is of the form } \Diamond^{\geq n}\phi_j \end{cases}$$

- Each $\mathrm{AGG}_i$ is (pointwise) sum.

It is not difficult to see that $\mathrm{COM}_i$ can be implemented by a two-layer feed-forward network using the ReLU activation function. In particular, $\max\{x_j, x_m\}$ can be expressed as $\mathrm{ReLU}(x_j + x_m - 1)$ and "1 if $y_j \geq n$, 0 otherwise" can be expressed as $\mathrm{ReLU}(1 - \mathrm{ReLU}(n - y_i))$.

It can be shown by a straightforward induction on $d$, that, for all $d \geq 0$, for all $\phi \in \Phi$ of operator depth $d$, and for all $i > d$, $\mathrm{emb}_G^i(v)(j)$ is 1 if $G, v \models \phi_j$ and 0 otherwise.

In order to turn $\mathcal{A}$ into a classifier, finally, we extend $\mathrm{COM}_L$ with one additional linear layer of dimensionality $(k, 1)$ that takes a vector $(x_1, \ldots, x_k)$ and outputs $x_i$, where $\phi_i = \phi$. $\qquad\square$

**Proposition 2.4.** *Let $\mathcal{A}$ be a $(D, D')$-GNN with $D = |P|$, let $N > 0$, and let*

$$X = \{\mathrm{run}_{\mathcal{A}}(G, \mathrm{emb}_G)(v) \mid G = (V, E, \mathrm{lab}) \text{ is a graph of degree at most } N \text{ and } v \in V\}.$$

*In other words $X \subseteq \mathbb{R}^{D'}$ is the set of all node embeddings that $\mathcal{A}$ can produce when run on a graph of degree at most $N$. Then $X$ is a finite set, and for each $\mathbf{x} \in X$, there is a* GML*-formula $\phi_{\mathbf{x}}$ such that for every pointed graph $(G, v)$ of degree at most $N$, it holds that $G, v \models \phi_{\mathbf{x}}$ iff $\mathrm{run}_{\mathcal{A}}(G, \mathrm{emb}_G)(v) = \mathbf{x}$. In particular, for each* GNN*-classifier $\mathsf{cls}_{\mathcal{A}}$ there is a* GML*-formula $\phi$ such that $\mathsf{cls}_{\phi}(G, v) = \mathsf{cls}_{\mathcal{A}}(G, v)$ for all pointed graphs $(G, v)$ of degree at most $N$.*

*Proof.* Let $\mathcal{A} = ((\mathrm{COM}_i)_{i=1\ldots L}, (\mathrm{AGG}_i)_{i=1\ldots L})$. By definition, $\mathrm{run}_{\mathcal{A}}(G, \mathrm{emb}_G)(v)$ is equal to $\mathrm{emb}_G^L(v)$ where, for each $i = 0 \ldots L$, $\mathrm{emb}_G^i : V \to \mathbb{R}^{D_i}$ is given by

- $\mathrm{emb}_G^0 = \mathrm{emb}_G$

- $\mathrm{emb}_G^i = \{v : \mathrm{COM}_i(\mathrm{emb}_G^{i-1}(v) \oplus \mathrm{AGG}_i(\{\!\{\mathrm{emb}_G^{i-1}(u) \mid (v, u) \in E\}\!\})) \mid v \in V\}$ for $i > 0$

The main statement can therefore be restated as saying that the set

$$X_L = \{\mathrm{emb}_G^L(v) \mid G = (V, E, \mathrm{lab}) \text{ is a graph of degree at most } N \text{ and } v \in V\}$$

is finite and there is a defining GML-formula $\phi_{\mathbf{x}}$ for each $\mathbf{x} \in X_L$ (over pointed graphs of degree at most $N$). We proceed by induction on $L$.

For $L = 0$, $\mathrm{emb}_G^L$ equals $\mathrm{emb}_G$. In this case, $X$ is equal to the set of all multi-hot encodings, i.e., $X = \{0, 1\}^D$, and for every $\mathbf{x} = (x_1, \ldots, x_D) \in X_L$, we can simply choose $\phi_{\mathbf{x}} = \alpha_1 \wedge \cdots \wedge \alpha_D$ where $\alpha_i = p_i$ if $x_i = 1$ and $\alpha_i = \neg p_i$ if $x_i = 0$.

Next, let $L > 0$. By induction hypothesis, the claim holds for $X_{L-1}$. Let us consider pairs $(\mathbf{z}, Z)$ where $\mathbf{z} \in X_{L-1}$ and $Z$ is a multiset of vectors belonging to $X_{L-1}$, such that $Z$ has cardinality at most $N$. Note that there are at most finitely many such pairs. For any such pair let $\phi_{(\mathbf{z},Z)}$ be the GML-formula

$$\phi_{\mathbf{z}} \wedge \bigwedge_{\mathbf{u} \text{ occurs in } Z \text{ with cardinality } n} (\Diamond^{\geq n} \phi_{\mathbf{u}} \wedge \neg \Diamond^{\geq n+1} \phi_{\mathbf{u}})$$

We now define our formula $\phi_{\mathbf{x}}$ as the disjunction of $\phi_{(\mathbf{z},Z)}$ for all pairs $(\mathbf{z}, Z)$ for which it holds that $\text{COM}_L(\mathbf{z} \oplus \text{AGG}_L(Z)) = \mathbf{x}$. It follows from the construction that $(G, v) \models \phi_{\mathbf{x}}$ if and only if $\text{emb}_G^L(v) = \mathbf{x}$. $\qquad \square$

# B  Proofs of logical characterizations in sections 3 and 4

We prove several lemmas building up to a uniform translation from HES-GNN-$(d,r)$ to $\text{GML}(\downarrow_{W^r}^d)$ (theorem B.5) and a converse uniform translation for graphs with bounded degree (theorem B.6). The logical characterizations in theorems 3.4 and 4.1 follow with small additions or adjustments to the proofs. We will use $\Diamond_k$ for $\underbrace{\Diamond\Diamond \ldots \Diamond}_{k}$.

**Proposition B.1.** *Every $\text{GML}(\downarrow, @)$-sentence is equivalent to a $\text{GML}(\downarrow)$-sentence of the same $\downarrow$-nesting-depth over unordered graphs.*

*Proof.* Let $\phi$ be a $\text{GML}(\downarrow, @)$-sentence. Let $n$ be the modal depth of $\phi$, that is, the maximal nesting depth of the modal operators in the sentence, and let $\phi'$ be the sentence obtained from $\phi$ by replacing every subformula of the form $@_x \psi$ by

$$\bigvee_{i=0\ldots 2n} (\underbrace{\Diamond \cdots \Diamond}_{\text{length } i}(x \wedge \psi))$$

Clearly, $\phi'$ has the same $\downarrow$-nesting-depth as $\phi$. A straightforward induction proof shows that $\phi$ and $\phi'$ are equivalent on unordered graphs. Intuitively, this is because all variables in $\phi$ are bound, and $2n$ bounds the distance between nodes in the subgraph that the formula $\phi$ can "see". $\qquad \square$

**Proposition B.2** (Canonical Form). *Let $Nd_\phi^W(\psi)$ be the number of $W$ operators in $\phi$ that have $\psi$ in their scope. For each $\text{GML}(\downarrow_{W^r}^d)$ sentence $\phi$ there exists an equivalent $\text{GML}(\downarrow_{W^r}^d)$ sentence $\psi$ such that for every subformula of the form $\downarrow x_i W^r \psi'$, $i = d + 1 - Nd_\psi^W(\psi')$. We call such $\psi$ canonical.*

*Proof.* We assume the proposition holds for $d$ and apply induction over formula construction in $\text{GML}(\downarrow_{W^r}^{d+1})$. Let $\phi_1 \in \text{GML}(\downarrow_{W^r}^d)$, and $\phi = \downarrow x_i.W^r \phi_1$. Let $\psi_1$ be the canonical form of $\phi_1$ and construct $\psi_1^*$ substituting free occurrences of $x_i$ in $\psi_1$ by $x_{d+1}$. Then:

$$\psi = \downarrow x_{d+1}.W^r \psi_1^*$$

Clearly $\psi \equiv \phi$, and for every subformula $\downarrow x_j. W^r \psi'$ in $\psi$ we have $j = d + 2 - Nd_\psi^W(\psi')$. Canonical form is maintained under construction with the other connectives $(\neg, \wedge, \Diamond^{>k})$. $\qquad \square$

**Lemma B.3.** *Given a finite tuple $(\mathcal{A}^1, \ldots \mathcal{A}^m)$ where for $1 \leq j \leq m$, $\mathcal{A}^j$ is a $(D, D')$-GNN, there exists a $(D, m \cdot D')$-GNN $\mathcal{A}$ such that $\text{run}_{\mathcal{A}}(G)(u) = \bigoplus_{1 \leq j \leq m} (\text{run}_{\mathcal{A}^j}(G)(u))$. Moreover, if all $\mathcal{A}^j$ use the same pointwise aggregation function, then $\mathcal{A}$ uses the same aggregation function, and if all $\mathcal{A}^j$ use ReLU-FFNNs as combination functions, the same holds for $\mathcal{A}$.*

*Proof.* For each $j$ let $\mathcal{A}^j = ((\text{COM}_i^j)_{i=1,\ldots,L_j}, (\text{AGG}_i^j)_{i=1,\ldots,L_j})$. Without loss of generality all $\mathcal{A}^j$ have the same number of layers $L'$. Let $L = L' + 1$. We construct $\mathcal{A} = ((\text{COM}_i)_{i=1,\ldots,L}, (\text{AGG}_i)_{i=1,\ldots,L})$. The first layer copies each embedding $m$ times, i.e. for $v, v' \in \mathbb{R}^D$: $\text{COM}_1(v \oplus v') = \bigoplus_{1 \leq j \leq m} (v)$.

Given combination functions $\text{COM}^j$ let $\underset{1\leq j\leq m}{\text{COM}}$ apply each $\text{COM}_j$ on the associated subspace:

$$\underset{1\leq j\leq m}{\text{COM}} (v_1 \oplus \ldots v_m) \oplus (v'_1 \oplus \ldots v'_m) = \bigoplus_{1\leq j\leq m} (\text{COM}^j(v_j \oplus v'_j))$$

Similarly, given aggregation functions $\text{AGG}^j$ let $\underset{1\leq j\leq m}{\text{AGG}}$ behave as:

$$\underset{1\leq j\leq m}{\text{AGG}} (M) = \bigoplus_{1\leq j\leq m} (\text{AGG}^j(M_j))$$

Here, for $\text{AGG}_j : \mathcal{M}(\mathbb{R}^{D_j}) \to \mathbb{R}^{D_j}$, $M$ is a multiset of embeddings in $\mathbb{R}^{\sum_{1\leq j\leq m} D_j}$ and $M_j$ is the multiset of embeddings in $\mathbb{R}^{D_j}$, obtained by restricting each embedding in $M$ to indices $[\sum_{j'<j} D_{j'}, \sum_{j'\leq j} D_{j'}]$. Note that if each $\text{AGG}_j$ is the same point-wise aggregation function (e.g. sum), then $\underset{1\leq j\leq m}{\text{AGG}}$ is the same point-wise aggregation function on the concatenated space.

We define the remaining layers of $\mathcal{A}$ as:

$$\forall l > 1 : \text{AGG}_l = \underset{1\leq j\leq m}{\text{AGG}_{l-1}}$$

$$\forall l > 1 : \text{COM}_l = \underset{1\leq j\leq m}{\text{COM}_{l-1}}$$

$\square$

**Corollary B.4.** *Given a finite tuple $(\mathcal{A}^1, \ldots \mathcal{A}^m)$ where for $1 \leq j \leq m$, $\mathcal{A}^j$ is a $(D, D')$-HES-GNN with nesting depth $d$ and radius $r$, there exists a $(D, m \cdot D')$-HES-GNN $\mathcal{A}$ with nesting depth $d$ and radius $r$ such that $\text{run}_\mathcal{A}(G)(u) = \bigoplus_{1\leq j\leq m} (\text{run}_{\mathcal{A}^j}(G)(u))$. Moreover, if all $\mathcal{A}^j$ use the same pointwise aggregation function, then $\mathcal{A}$ uses the same aggregation function, and if all $\mathcal{A}^j$ use ReLU-FFNNs as combination functions, the same holds for $\mathcal{A}$.*

*Proof.* For $d = 0$ the claim follows from the above lemma. For $d > 0$ and $\mathcal{A}^j = (\mathcal{B}^j, \mathcal{C}^j)$, construct $\mathcal{A} = (\mathcal{B}, \mathcal{C})$ where $\text{run}_\mathcal{B}(G)(u) = \bigoplus_{1\leq j\leq m} (\text{run}_{\mathcal{B}^j}(G)(u))$.

$\mathcal{C}$ follows the construction of the above lemma, with the exception that the first combination function does not copy the complete vertex embedding $m$ times. Instead, $\text{COM}_1$ now receives a vertex embedding $v \oplus \bigoplus_{1\leq j\leq m} (v_j)$, where $v$ is the original vertex embedding and $v_j$ is the output of $\mathcal{B}^j$, and produces output $\bigoplus_{1\leq j\leq m} (v \oplus v_j)$. $\square$

**Theorem B.5.** *Given a finite tuple $(\phi_1, \ldots, \phi_m)$ where for $1 \leq j \leq m$, $\phi_j$ is a $\text{GML}(\downarrow^d_{W^r})$ sentence with propositions in $P$, there exists a $(|P|, m)$-HE-GNN $\mathcal{A}$ with nesting depth $d$ and radius $r$ such that for all $G$ labeled with $P$ $\text{run}_\mathcal{A}(G)(u) = \bigoplus_{1\leq j\leq m} (\text{cls}_{\phi_j}(G, u))$.*

*Proof.* For $d = 0$, following the uniform translation from GML to GNN of proposition 2.3, there exist a GNN $\mathcal{A}_j$ such that $\text{run}_{\mathcal{A}_j} = \text{cls}_{\phi_j}$. By lemma B.3 there then exists a GNN $\mathcal{A}$ that produces the concatenation of outputs for each $\mathcal{A}_j$.

We apply induction over $d$. It suffices to show that any single sentence $\phi$ in $\text{GML}(\downarrow^d_{W^r})$ is implemented by a single HE-GNN $\mathcal{A}$, since we can construct a HE-GNN with concatenated outputs following corollary B.4.

By lemma B.2 we can assume $\phi$ is canonical, hence for every maximal subformula $\downarrow x_i.W^r(\psi)$ in $\phi$, $i = d$. Let $\downarrow x_d.W^r\psi_1, \ldots, \downarrow x_d.W^r\psi_k$ be all such maximal subformulas. By the semantics of $\text{GML}(\downarrow^d_{W^r})$, for all $1 \leq j \leq k$:

$$G, v \models \downarrow x_d.W^r(\psi_j) \text{ iff}$$
$$G^r_v[x_d \mapsto v], v \models \psi_j$$

Now when $x_d$ is treated as a binary node feature, $\psi_j$ is a sentence in $\mathrm{GML}(\downarrow^d_{W^r})$. There thus exists a HE-GNN $\mathcal{B}_j$ such that

$$\mathrm{run}_{\mathcal{B}_j}(G^r_v[x_d \mapsto v])(v) = \mathsf{cls}_{\psi_j}(G^r_v[x_d \mapsto v], v)$$
$$= \mathsf{cls}_{\downarrow_{x_d}.W^r(\psi_j)}(G, v)$$

We construct a single HE-GNN $\mathcal{B}$ with nesting depth $d-1$ that outputs the concatenated outputs for all such $\mathcal{B}_j$, following corollary B.4:

$$\mathrm{run}_{\mathcal{B}}(G^r_v[x_d \mapsto v])(v) = \bigoplus_{1 \leq j \leq m} \mathsf{cls}_{\downarrow x_d.W^r(\psi_j)}(G, v)$$

Now construct $\phi^*$ from $\phi$ by substituting each $\downarrow x_d.W^r\psi_j$ with a proposition $q_j$. Given $G = (V, E, \mathrm{lab})$, let $G^* = (V, E, \mathrm{lab}^*)$, where $\mathrm{lab}^*$ extends lab so that $q_j \in \mathrm{lab}^*(u)$ iff $G, u \models\downarrow x_d.W^r\psi_j$. Then:

$$\mathsf{cls}_\phi(G, v) = \mathsf{cls}_{\phi^*}(G^*, v)$$

Note that $\phi^* \in \mathrm{GML}$. Hence by proposition 2.3 there exists a GNN $\mathcal{C}$ such that for $\mathcal{A} = (\mathcal{B}, \mathcal{C})$:

$$\mathsf{cls}_{\phi^*}(G^*, v) = \mathrm{run}_{\mathcal{C}}(G^*, v)$$
$$= \mathrm{run}_{\mathcal{A}}(G, v)$$

$\square$

**Theorem B.6.** *Let $\mathcal{A}$ be a $(D, D')$-HES-GNN with $D = |P|$, nesting depth $d$, radius $r$. Let $N \geq 0$ and*

$$X = \{\mathrm{run}_{\mathcal{A}}(G, \mathrm{emb}_G)(v) \,|\, G = (V, E, \mathrm{lab}) \text{ is a graph of degree at most } N \text{ and } v \in V\}$$

*Then $X$ is a finite set, and for each $\mathbf{x} \in X$ there is a $\mathrm{GML}(\downarrow^d_{W^r})$ sentence $\phi_{\mathbf{x}}$ such that for every pointed graph $(G, v)$ of degree at most $N$ it holds that $G, v \models \phi_{\mathbf{x}}$ iff $\mathrm{run}_{\mathcal{A}}(G)(v) = \mathbf{x}$. In particular, for each HES-GNN-(d,r) classifier $\mathsf{cls}_{\mathcal{A}}$ there is a $\mathrm{GML}(\downarrow^d_{W^r})$ sentence $\phi$ such that $\mathsf{cls}_\phi(G, v) = \mathsf{cls}_{\mathcal{A}}(G, v)$ for all pointed graphs $(G, v)$ of degree at most $N$.*

*Proof.* We apply induction over $d$. The case $d = 0$ reduces to the translation from GNN to GML of proposition 2.4. Let $\mathcal{A} = (\mathcal{B}, \mathcal{C})$ be a HES-GNN with depth $d$ and radius $r$. $X$ is finite since $\mathcal{B}$ has finitely many output embeddings by the induction hypothesis, and $\mathcal{C}$ produces finitely many output features on graphs with finitely many input embeddings as shown in the proof of proposition 2.4.

Let $\mathrm{emb}^u = \{\mathrm{emb}(u') \oplus \delta_{uu'} \,|\, u' \in V\}$, then for each $\mathbf{x} \in X$:

$$\mathrm{run}_{\mathcal{A}}(G)(v) = \mathbf{x} \text{ iff}$$
$$\mathrm{run}_{\mathcal{C}}(G, \{\mathrm{emb}_G(u) \oplus \mathrm{run}_{\mathcal{B}}(G^r_u, \mathrm{emb}^u) | u \in V\})(v) = \mathbf{x} \text{ iff}$$
$$\mathrm{run}_{\mathcal{C}}(G^*)(v) = \mathbf{x}$$

Where given $G = (V, E, \mathrm{lab})$, $G^* = (V, E, \mathrm{lab}^*)$ and $\mathrm{lab}^*$ extends lab with propositions for all output features of $\mathcal{B}$. Specifically, let $Y_{\mathcal{B}} = \{\mathbf{y_1}, \ldots \mathbf{y}_{|Y_{\mathcal{B}}|}\}$ be this finite set of output features and introduce $q_1 \ldots q_{|Y_{\mathcal{B}}|}$ such that $q_j \in \mathrm{lab}^*(u)$ if and only if $\mathrm{run}_{\mathcal{B}}(G^r_u, \mathrm{emb}^u)(u) = \mathbf{y_j}$.

By proposition 2.4 there exists a sentence $\xi$ in GML such that:

$$G^*, v \models \xi \text{ iff } \mathrm{run}_{\mathcal{C}}(G^*)(v) = \mathbf{x}$$

Then let $\phi_{\mathbf{x}} = \xi[\downarrow x_d.W^r\phi_{\mathbf{y_1}}, \ldots, \downarrow x_d.W^r\phi_{\mathbf{y}_{|Y_{\mathcal{B}}|}}/q_1, \ldots q_{|Y_{\mathcal{B}}|}]$.

Here all $\downarrow x_d.W^r(\phi_{\mathbf{y_i}})$ are sentences in $\mathrm{GML}(\downarrow^d_{W^r})$ by the induction hypothesis, so that the same holds for $\phi_{\mathbf{x}}$. $\square$

Small adaptations can be made to the proofs of theorems B.5 and B.6 to show $\rho(\mathrm{HE\text{-}GNN\text{-}}d) = \rho(\mathrm{GML}(\downarrow^d))$, where now all operators $W^r$ are removed and no subgraph restrictions are applied. Since $\mathrm{HE\text{-}GNN} = \cup_{d \geq 0}\mathrm{HE\text{-}GNN\text{-}}d$ and $\mathrm{GML}(\downarrow) = \cup_{d \geq 0}\mathrm{GML}(\downarrow^d)$ it follows that $\rho(\mathrm{HE\text{-}GNN}) = \rho(\mathrm{GML}(\downarrow))$. That is:

**Theorem 3.4.** $\rho(\text{HE-GNN}) = \rho(\text{GML}(\downarrow))$. *Moreover, for $d \geq 0$, $\rho(\text{HE-GNN-}d) = \rho(\text{GML}(\downarrow^d))$.*

**Lemma B.7.** *Let $r \geq 0$, then $\rho(\text{GML}(\downarrow^d)) \subseteq \rho(\text{GML}(\downarrow^d_{W^r}))$*

*Proof.* Let $\phi \in \text{GML}(\downarrow^d_{W^r})$. We define sentence $\xi$, which only holds at vertices of distance $\leq r$ to $x_i$:

$$\xi = x_i \vee \bigvee_{1 \leq j \leq r} \Diamond_j x_i$$

Now take a minimal subformula of the form $\downarrow x_i . W^r \psi$ in $\phi$. We substitute this for an equivalent subformula $\downarrow x_i . \psi'$, where $\Diamond^{\geq k} \tau$ in $\psi$ is replaced by $\Diamond^{\geq k}(\xi \wedge \tau)$ to produce $\psi'$.

Applying this transformation recursively to subformulas in $\phi$ yields an equivalent $\text{GML}(\downarrow^d)$ sentence. $\qquad \square$

**Theorem 4.1.**

1. $\rho(\text{HES-GNN}) = \rho(\text{GML}(\downarrow_W)) = \rho(\text{GML}(\downarrow)) = \rho(\text{HE-GNN})$.

2. $\rho(\text{HES-GNN-}r) = \rho(\text{GML}(\downarrow_{W^r}))$. *Moreover, $\rho(\text{HES-GNN-}(d,r)) = \rho(\text{GML}(\downarrow^d_{W^r}))$.*

*Proof.* Theorems B.5 and B.6 yield $\rho(\text{HES-GNN-}(d,r)) = \rho(\text{GML}(\downarrow^d_{W^r}))$ for all $d, r \geq 0$.

Since $\text{HE-GNN-}r = \cup_{d \geq 0}\text{HES-GNN-}(d,r)$ and $\text{GML}(\downarrow_{W^r}) = \cup_{d \geq 0}\text{GML}(\downarrow^d_{W^r})$ point (2) follows.

Similarly, taking the union over all $r \geq 0$ yields $\rho(\text{HES-GNN}) = \rho(\text{GML}(\downarrow_W))$. Finally, to see that $\rho(\text{GML}(\downarrow_W)) = \rho(\text{GML}(\downarrow))$ note that when two pointed graphs are separated by a sentence in $\text{GML}(\downarrow)$ they are also separated by a $\text{GML}(\downarrow_{W^r})$ sentence where $r$ is the size of the largest of the two graphs. Using lemma B.7 then $\rho(\text{GML}(\downarrow_W)) = \rho(\text{GML}(\downarrow))$. $\qquad \square$

## C    Proofs of hierarchy results in section 4

**Theorem 4.2.** *For $d \geq 1, r \geq 0$, $\rho(\text{HES-GNN-}(d,r+1)) \subsetneq \rho(\text{HES-GNN-}(d,r))$.*

The construction of lemma B.7 also shows $\rho(\text{GML}(\downarrow^d_{W^{r+1}})) \subseteq \rho(\text{GML}(\downarrow^d_{W^r}))$ for $d \geq 1, r \geq 0$. Using the logical characterization of theorem 4.1 then $\rho(\text{HES-GNN-}(d,r+1)) \subseteq \rho(\text{HES-GNN-}(d,r))$.

Now to show $\rho(\text{HES-GNN-}(d,r+1)) \neq \rho(\text{HES-GNN-}(d,r))$, consider graphs $G_1 = C_{4r+6}$ and $G_2$ consisting of two disjoint cycles $C_{2r+3}$, where all vertices are labeled with the empty set.

Let $v_1, v_2$ be nodes in $G_1, G_2$ and $\phi = \downarrow x_i . W^{r+1}(\Diamond_{r+1}(\neg \Diamond^{\geq 2} \top))$. Then:

$$G_1, v_1 \models \phi$$
$$G_2, v_2 \not\models \phi$$

Again applying theorem 4.1 $(G_1, v_1)$ and $(G_2, v_2)$ can be separated by HES-GNN-$(d,r+1)$ for $d \geq 1$.

However, the marked induced subgraphs with radius $r$ around $v_1, v_2$ are isomorphic and hence cannot be separated by any HE-GNN. Since $(G_1, v_1), (G_2, v_2)$ are further indistinguishable by WL:

$$\forall d \geq 0 : ((G_1, v_1), (G_2, v_2)) \in \rho(\text{HES-GNN-}(d,r)))$$

**Theorem 4.3.** *For $d \geq 0$ and $r \geq 3$, $\rho(\text{HES-GNN-}(d+1,r)) \subsetneq \rho(\text{HES-GNN-}(d,r))$*

This follows from

$$\rho((d+2)\text{-GNN}) \subseteq \rho(\text{HE-GNN-}d) \subseteq \rho(\text{HES-GNN-}(d,r)) \qquad \text{Theorem 5.11}$$
$$\rho((d+2)\text{-GNN}) \not\subseteq \rho(\text{HES-GNN-}(d+1,r)) \qquad \text{Theorem 5.12}$$

# D   Proofs of relations with other models in section 5

## 5.1. Individualization-Refinement

**Proposition D.1** (([38, 50]). *Let $n, D \in \mathbb{N}$. Then there exists a GNN $\mathcal{A}^s$ such that for all featured graphs $(G, \mathrm{emb}), (G', \mathrm{emb}')$ of size $\leq n$ and with embeddings in $\mathbb{R}^D$, if $\mathrm{WL}(G, n)(v) \neq \mathrm{WL}(G', n)(v')$ then $\mathrm{run}_{\mathcal{A}^s}(G)(v) \neq \mathrm{run}_{\mathcal{A}^s}(G')(v')$. We call such a GNN sufficiently separating.*

**Lemma D.2.** *Let $n, d, D \in \mathbb{N}$. Then there exists a sufficiently separating HE-GNN $\mathcal{A}^s$ with nesting depth $d$ such that for all featured graphs $(G, \mathrm{emb}), (G', \mathrm{emb}')$ of size $\leq n$ and with embeddings in $\mathbb{R}^D$, if $((G, v), (G', v')) \notin \rho(\text{HE-GNN-}d)$ then $\mathrm{run}_{\mathcal{A}^s}(G)(v) \neq \mathrm{run}_{\mathcal{A}^s}(G')(v')$, we call such a HE-GNN sufficiently separating.*

*Proof.* When $d = 0$ this reduces to proposition D.1. Suppose $d > 0$. Let $\mathcal{B}^s$ be a sufficiently separating HE-GNN for $n, d-1, D+1$, and let $\mathcal{C}^s$ be a sufficiently separating GNN for $n, D'$, where $D'$ is the output dimension of $\mathcal{B}^s$. We define $\mathcal{A}^s = (\mathcal{B}^s, \mathcal{C}^s)$.

Now suppose a HE-GNN $\mathcal{A}' = (\mathcal{B}', \mathcal{C}')$ with nesting depth $d$ separates $(G, v)$ from $(G', v')$. Since $\mathcal{C}^s$ is sufficiently separating, $(\mathcal{B}', \mathcal{C}^s)$ is at least as separating as $(\mathcal{B}', \mathcal{C}')$. Since $\mathcal{C}^s$ has the separating power of WL up to size $n$, and since $\mathcal{B}^s$ has at least the separating power of $\mathcal{B}'$, $(\mathcal{B}^s, \mathcal{C}^s)$ is at least as separating as $(\mathcal{B}', \mathcal{C}^s)$.  $\square$

**Notation** Assume an injective map $f$ from the infinite set of colors $\mathcal{C}$ to an infinite set of binary node labels (propositions) $\mathcal{P}$. Given a graph $G = (V, E, \mathrm{lab})$ we write $G_{\mathrm{WL}}$ for $G = (V, E, \mathrm{lab}')$, where $\mathrm{lab}'$ is obtained by first computing the coloring $\mathrm{col} = \mathrm{WL}(G, |G|)$ and then labeling each node $v \in V$ with $f(\mathrm{col}(v))$.

**Lemma D.3.** *Let $(G, v), (G', v')$ be pointed graphs such that $|G| = |G'|$. If $((G, v), (G', v')) \in \rho(\text{HE-GNN-}d)$ then $((G_{\mathrm{WL}}, v), (G'_{\mathrm{WL}}, v')) \in \rho(\text{HE-GNN-}d)$.*

*Proof.* We apply induction over $d$. For $d = 0$ note that $\rho(\text{HE-GNN-0}) = \rho(\text{GNN}) = \rho(\text{WL})$. Let $(G, v), (G', v')$ be of size $n$, with labels in $P$, such that $((G, v), (G', v')) \in \rho(\text{HE-GNN-}d)$ for $d > 0$.

By lemma D.2 there exists a sufficiently separating HE-GNN $\mathcal{A}^s = (\mathcal{B}^s, \mathcal{C}^s)$. We show $(G_{\mathrm{WL}}, v), (G'_{\mathrm{WL}}, v') \in \rho(\mathsf{cls}_{\mathcal{A}^s})$.

Suppose for nodes $u$ in $G$ and $u'$ in $G'$, are given the same output by $\mathcal{B}^s$ after individualizing $u, u'$. By the induction hypothesis, the same holds for $u$ in $G_{\mathrm{WL}}$ and $u'$ in $G'_{\mathrm{WL}}$. Since $\mathcal{C}^s$ is sufficiently separating and since the input embeddings given by $\mathcal{C}^s$ are not more separating for $G_{\mathrm{WL}}, G'_{\mathrm{WL}}$ than for $G, G'$, it follows that $(G_{\mathrm{WL}}, v), (G'_{\mathrm{WL}}, v') \in \rho(\text{HE-GNN-}d)$.  $\square$

We apply the following lemma from Zhang et al. [52]

**Lemma D.4.** *Let $G = (V, E, \mathrm{lab})$, $G' = (V', E', \mathrm{lab}')$ be finite connected graphs with $v \in V, v' \in V'$ uniquely marked. Then:*

$$\mathrm{WL}(G)(v) = \mathrm{WL}(G')(v') \text{ iff } \{\!\{ \mathrm{WL}(G)(u) \mid u \in V \}\!\} = \{\!\{ \mathrm{WL}(G')(u') \mid u' \in V' \}\!\}$$

**Theorem 5.3.** *For $d \geq 0$, if $G \not\equiv_{\text{WL-IR-}d} G'$, then $G \not\equiv_{\text{HE-GNN-}d} G'$*

*Proof.* We apply induction over $d$. At $d = 0$, the claim follows since $\rho(\text{GNN}) = \rho(\text{WL})$. Suppose for graphs $G, G'$ with labels in $P$ and $d > 0$, $G \equiv_{\text{HE-GNN-}(d)} G'$, then $|G| = |G'|$. We show $G \equiv_{\text{WL-IR-}(d)} G'$.

By lemma D.2 there exists a sufficiently separating HE-GNN $\mathcal{A}^s = (\mathcal{B}^s, \mathcal{C}^s)$. Since:

$$\{\!\{ \mathrm{run}_{\mathcal{A}^s}(G, v) \mid v \in V \}\!\} = \{\!\{ \mathrm{run}_{\mathcal{A}^s}(G', v') \mid v' \in V' \}\!\}$$

there is a bijection $f$ between the two multisets, such that for all $u \in V$ and for unique proposition $q$:

$$\mathrm{run}_{\mathcal{B}^s}(G_{[q \mapsto u]})(u) = \mathrm{run}_{\mathcal{B}^s}(G'_{[q \mapsto f(u)]})(f(u))$$

Let $C^u, C^{f(u)}$ be the connected components containing $u$ and $f(u)$. By lemma D.4:

$$\{\!\{ \mathrm{run}_{\mathcal{B}^s}((C^u_{[q \mapsto u]})_{\mathrm{WL}})(w) \mid w \in C^u \}\!\} = \{\!\{ \mathrm{run}_{\mathcal{B}^s}((C^{f(u)}_{[q \mapsto f(u)]})_{\mathrm{WL}})(w) \mid w \in C^{f(u)} \}\!\}$$

Since these connected components then also obtain the same multiset of output embeddings without a unique marking, this equality extends to the full graphs $G, G'$:

$$\{\!\{\mathrm{run}_{\mathcal{B}^s}(G_{[q\mapsto u]})(w) \mid w \in V\}\!\} = \{\!\{\mathrm{run}_{\mathcal{B}^s}(G'_{[q\mapsto f(u)]})(w) \mid w \in V'\}\!\}$$

It follows that $G, G'$ are indistinguishable by WL-IR-$(d-1)$ after marking $u$, and $f(u)$ and applying WL:

$$\{\!\{\mathrm{run}_{\mathcal{B}^s}((G_{[q\mapsto u]})_{\mathrm{WL}})(w) \mid w \in V\}\!\} = \{\!\{\mathrm{run}_{\mathcal{B}^s}((G'_{[q\mapsto f(u)]})_{\mathrm{WL}})(w) \mid w \in V'\}\!\} \quad \text{(Lemma D.3)}$$

$$(G_{[q\mapsto u]})_{\mathrm{WL}} \equiv_{\text{HE-GNN-}(d-1)} (G'_{[q\mapsto f(u)]})_{\mathrm{WL}} \quad \text{(Lemma D.2)}$$

$$(G_{[q\mapsto u]})_{\mathrm{WL}} \equiv_{\text{WL-IR-}(d-1)} (G'_{[q\mapsto f(u)]})_{\mathrm{WL}} \quad \text{(Induction hypothesis)}$$

Since WL-IR$(G, d)$ obtains the same depth $d-1$ subtrees as WL-IR$(G', d)$, $G \equiv_{\text{WL-IR-d}} G'$. $\qquad\square$

**Theorem 5.4.** *Let* $(G, v), (G', v')$ *be connected pointed graphs and let* $d \geq 0$. *If* $G \not\equiv_{\text{WL-IR-}d} G'$, *there exists a depth* $d+1$ *HE-GNN* $\mathcal{A}$ *such that* $\mathsf{cls}_{\mathcal{A}}(G, v) \neq \mathsf{cls}_{\mathcal{A}}(G', v')$.

*Proof.* Let $d \geq 0$. Suppose $(G, v) \equiv_{\text{HE-GNN-}d+1} (G', v')$. We show $G \equiv_{\text{WL-IR-}d} G'$.

For a sufficiently separating HE-GNN $\mathcal{A}^s = (\mathcal{B}^s, \mathcal{C}^s)$ with nesting depth $d+1$:

$$\mathrm{run}_{\mathcal{A}^s}(G)(v) = \mathrm{run}_{\mathcal{A}^s}(G')(v')$$

Thus, for unique label $q$:

$$\mathrm{run}_{\mathcal{B}^s}(G_{[q\mapsto v]})(v) = \mathrm{run}_{\mathcal{B}^s}(G'_{[q\mapsto v']})(v')$$

Then since $G, G'$ are connected:

$$\{\!\{\mathrm{run}_{\mathcal{B}^s}(G_{[q\mapsto v]})(u) \mid u \in V\}\!\} = \{\!\{\mathrm{run}_{\mathcal{B}^s}(G'_{[q\mapsto v']})(u') \mid u' \in V'\}\!\} \quad \text{(Lemma D.4)}$$

$$G \equiv_{\text{HE-GNN-}d} G' \quad \text{(Lemma D.2)}$$

$$G \equiv_{\text{WL-IR-}d} G' \quad \text{(Theorem 5.3)}$$

$$\square$$

Figure 5: Two graphs that are distinguished by a depth 1 HE-GNN, but not by WL-IR-1. All nodes in both graphs have an empty labeling.

**Theorem 5.5.** *There exist* $G, G'$ *such that* $G =_{\text{WL-IR-1}} G'$ *but* $G \neq_{\text{HE-GNN-1}} G'$

*Proof.* We use two graphs $G, G'$ introduced by Rattan and Seppelt [44], and shown in Figure 5. $G$ has a square $u_1, u_2, u_3, u_4$, where $u_1$ is connected to all nodes in two triangles, the same holds for $u_4$ and two different triangles, and $u_2, u_3$ are connected to all nodes in two distinct 6 cycles. $G'$ is constructed similarly with a central square $v_1, v_2, v_3, v_4$, where now $v_1$ and $v_3$ are each connected to two triangles and $v_2, v_4$ are connected to distinct 6 cycles.

To see $G =_{\text{WL-IR-1}} G'$ note firstly that the two graphs are WL equivalent. Furthermore, there is a bijection $f$ between nodes of $G$ and $G'$, where for each node $u$ in $G$, applying WL to $G$ after individualizing $u$ yields the same coloring as applying WL to $G'$ after individualizing $f(u)$. We let $f(u_1) = v_1, f(u_2) = v_2, f(u_3) = v_3, f(u_4) = v_4$, map every node in a triangle in $G$ to a node in a triangle in $G'$ and do the same for nodes in the 6 cycles of $G$ and $G'$. One can easily check that this gives the desired result so that $G =_{\text{WL-IR-1}} G'$.

We now show there is a GML($\downarrow^1_{W^2}$) formula $\psi$ that is satisfied by $u_1$ but not by any node in $G'$.

$$\phi = \Diamond^{\geq 8}(\top) \wedge \Diamond(\downarrow x. W^2(\neg \Diamond^{\geq 8}(\top) \wedge \Diamond(\neg \Diamond^{\geq 8}(\top) \wedge \Diamond(\neg \Diamond^{\geq 8}(\top) \wedge \Diamond x))))$$
$$\psi = \phi \wedge \neg \Diamond \phi$$

Here, $\phi$ expresses having degree at least $8$, and being connected to a node on a 3 cycle that doesn't pass a node with degree at least $8$. $u_1, u_4, v_1$ and $v_3$ are the only nodes in $G, G'$ that satisfy $\phi$. Thus $G, u_1 \models \psi$, while no node in $G'$ satisfies $\psi$. By the logical characterization in theorem 3.4 there exists a HE-GNN $\mathcal{A}$ such that $G \neq_{\text{cls}_{\mathcal{A}}} G'$.  $\square$

## 5.2. Homomorphism count enriched GNNs

The definitions of homomorphisms and homomorphism counts were omitted from the paper due to lack of space. They are as follows: a *homomorphism* from a pointed graph $(F, u)$ to a pointed graph $(G, v)$ is a map $h$ from the vertex set of $F$ to the vertex set of $G$ such that

1. for each edge $(w, w')$ of $F$, $(h(w), h(w'))$ is an edge of $G$,
2. for each vertex $w$ of $F$, $lab_F(w) \subseteq lab_G(h(w))$, and
3. $h(u) = v$.

Homomorphisms are defined similarly for unpointed graphs $G$, where we simply omit condition (ii). We use $\hom((F, u), (G, v))$ to denote the number of homomorphism from $(F, u)$ to $(G, v)$. In addition, if $h$ is a partial map from the vertex set of $F$ to the vertex set of $G$, then we denote by $\hom_h(F, G)$ the number of homomorphism that extend $h$. In particular, $\hom_{\{(u,v)\}}(F, G) = \hom((F, u), (G, v))$.

For a set of pointed graphs $\mathfrak{F} = \{(F_1, u_1), \ldots, (F_m, u_m)\}$ and a pointed graph $(G, v)$, we denote by $\hom(\mathfrak{F}, (G, v))$ the vector of homomorphism counts

$$\langle \hom((F_1, u_1), (G, v)), \ldots, \hom((F_m, u_m), (G, v)) \rangle$$

(assuming some ordering on the members of $\mathfrak{F}$).

**Theorem 5.6.** *Let $\mathfrak{F}$ be any finite set of rooted graphs each with at most $d$ nodes. Then there is a $(|P|, |P| + |\mathfrak{F}|)$-HE-GNN $\mathcal{A}$ of nesting depth $d$ such that, for all pointed graphs $(G, v)$,*

$$\text{run}_{\mathcal{A}}(G)(v) = \text{emb}_G(v) \oplus \hom(\mathfrak{F}, (G, v))$$

*The HE-GNN in question only uses Sum as aggregation and ReLu-FFNNs as combination functions.*

*Proof.* We consider the case where $\mathfrak{F}$ consists of a single rooted graph $(F, u)$ with $d$ nodes. The proof extends naturally to the case with multiple such rooted graphs.

We will prove the following stronger statement:

(*) For all sequences $\langle u_1, \ldots, u_k \rangle$ of distinct nodes of $F$ (with $k > 0$), there is a $(D + k, 1) -$ HE-GNN $\mathcal{A}$ of nesting depth $d - k$, such that, for all graph $G$ and maps $h : \{u_1, \ldots, u_k\} \to V_G$,

$$\text{run}_{\mathcal{A}}(G, \text{emb}_G^{+h})(h(u_k)) = \hom_h(F, G)$$

where $\text{emb}_G^{+h} = \{w : \text{emb}_G(w) \oplus \langle \delta_{wh(u_1)}, \ldots, \delta_{wh(u_k)} \rangle \mid w \in V_G\}$.

Observe that the special case of (*) with $k = 1$ and $u_1 = u$ yields a $(|P| + 1, 1)$-HE-GNN $\mathcal{B}$ of nesting depth $d - 1$ such that

$$\text{run}_{\mathcal{B}}(G, \text{emb}_G')(v) = \hom((F, u), (G, v))$$

where $\text{emb}_G' = \{w : \text{emb}_G(w) \oplus \langle \delta_{wh(u)} \rangle \mid w \in V_G\}$. Let $\mathcal{C}$ be the trivial $(|P|+1, |P|+1)$-GNN that implements the identity function. It then follows that $\mathcal{A} = (\mathcal{B}, \mathcal{C})$, which is a $(|P|, |P|+1)$-HE-GNN of nesting depth $d$, has the desired behavior.

It remains to prove (*). The proof proceeds by induction on the $d - k$. When $k = d$, the partial function $h$ is in fact a total function from the node set of $F$ to that of $G$. It is easy to implement a GNN that, in this case, outputs 1 if $h$ is a homomorphism and outputs 0 otherwise. Indeed, this can be done using only ReLU-FFNN combination functions and Sum aggregation, and using at most $|V_F|$ many rounds of message passing. We omit the details, as they are straightforward.

Next, let $0 < k < d$ and assume that (*) holds for $k + 1$. We will show that it then also holds for $k$. Since $k > 0$, there are nodes of $F$ that do not belong to the sequence $\langle u_1, \ldots, u_k \rangle$. It follows from this, by the connectedness of $F$, that there is an edge of $F$ connecting some $u_i$ (with $i \leq k$) to some $u' \notin \{u_1, \ldots, u_k\}$. As a basic fact about homomorphism counts, we have

$$\hom_h(F, G) = \sum_{v' \in V_G \text{ such that } (h(u_i), v') \in E_G} \hom_{h \cup \{(u', v')\}}(F, G)$$

We now apply induction hypothesis to $\langle u_1, \ldots, u_k, u' \rangle$, obtaining a $(|P| + k + 1, 1)$-HE-GNN $\mathcal{B}$. Let $\mathcal{C}$ be a $(|P| + k + 1, |P| + k + 1)$-GNN that performs one round of message passing using Sum aggregation and using the combination function

$$\text{COM}(\langle x_1, \ldots, x_{|P|+k}, x', z_1, \ldots, z_{|P|+k}, z' \rangle) = \langle x_1, \ldots, x_{|P|+k}, z' \rangle$$

(i.e. summing up the values in the $|P| + k + 1$-th position across all neighbors, and keeping the other values in the vector the same). Let $\mathcal{A} = (\mathcal{B}, \mathcal{C})$. It follows from the construction that

$$\text{run}_{\mathcal{A}}(G, \text{emb}_G^{+h})(h(u_i)) = \hom_h(F, G)$$

In other words, after running $\mathcal{A}$, "node $h(u_i)$ knows the answer". All that remains to complete the construction, is to "pass this information from $h(u_i)$ to $h(u_k)$. This can be done by augmenting $\mathcal{C}$ with $|V_F|$ more layers of message passing (because $h(u_i)$ and $h(u_k)$ are at most $|V_F|$ distance apart). We omit the details which are straightforward. $\square$

**Lemma D.5.** *For every pointed graph $(F, u)$ of ego-rank $n$, there is a witnessing dep-function (i.e., with maximum node rank $n$) such that*

1. *dep is well-founded, i.e, $v \neq dep^n(v)$ for all $v$ and $n \geq 1$.*

2. *for all nodes $v$, every connected component of the subgraph induced by $dep^{-1}(v)$ contains a neighbor of $v$. Equivalently, when $dep(w) = v$, there is a path from $w$ to $v$ passing only though nodes $w'$ with $dep(w') = v$.*

*Proof.*

1. If $dep$ is not well-founded, there is a cycle

$$v_1, v_2, v_3, \ldots, v_n$$

where $dep(v_i) = v_{i+1}$ for $i < n$ and $dep(v_n) = v_1$. Note that, in this case, $deps(v_1) = deps(v_2) = \ldots = deps(v_n) = \{v_1, \ldots, v_n\}$.

Fix such a cycle, and let $dep'$ be identical to $dep$ except that (i) $dep'(v_n) = \bot$, and (ii) for all $v \notin \{v_1, \ldots, v_n\}$, if $dep(v) \in \{v_1, \ldots, v_n\}$ then $dep'(v) = v_1$. Note that, in this way, $deps'(v) = deps(v)$ for all $v \notin \{v_1, \ldots, v_n\}$.

We claim that $dep'$ still satisfies the conditions given in the definition of ego-rank. Indeed,

- $dep'(u)$ is still $\bot$

- Let $(w, v) \in E$ be an edge. Then one of the following three cases holds:

  (a) $dep(w) = dep(v)$. Then the same holds for $dep'$, except possibly if $w \notin \{v_1, \ldots, v_n\}$, $dep(w) \in \{v_1, \ldots, v_n\}$, and $v \in \{v_1, \ldots, v_n\}$. However, in this case, we have that $dep'(w) = v_1$ and hence $v \in deps'(w) = \{v_1, \ldots, v_n\}$.

  (b) $w \in deps(v)$. It follows from the construction of $dep'$ that, for all $v \notin \{v_1, \ldots, v_n\}$, $deps'(v) = deps(v)$. Therefore, we only have to consider the case that $v \in \{v_1, \ldots, v_n\}$. If $w \in \{v_1, \ldots, v_n\}$, then we have either $w \in deps'(v)$ or $v \in deps'(w)$ (note that $v \neq w$). Otherwise, by construction, $dep'(w) = v_1$ and hence $v \in deps'(w)$.



Figure 6: Rooted $5 \times 2$ grid

(c) $v \in deps(w)$. This case is symmetric to the above.

- Finally, we must show that, for each $v \in V \cup \{\bot\}$, the subgraph induced by $dep'^{-1}(v)$ is acyclic. For each node $v \neq v_1$, we have that $dep'^{-1}(v) \subseteq dep^{-1}(v)$, and hence, since $dep^{-1}(v)$ is acyclic, so is $dep'^{-1}(v)$. Therefore, it remains only to consider $dep'^{-1}(\bot)$ and $dep'^{-1}(v_1)$.

  Suppose there were a cycle in the subgraph induced by $dep'^{-1}(\bot)$. This cycle must contain the node $v_1$, while all other nodes $u$ on the cycle satisfy $dep(u) = \bot$. However, it is easy to see that there can be no edge connecting $v_1$ to such a node $u$.

  Finally, suppose there were a cycle

  $$w_1, \ldots, w_k$$

  in the subgraph induced by $dep'^{-1}(v_1)$. If $dep(w_1) = dep(w_2) = \ldots = dep(w_k) = v_i$, then the subgraph induced by $dep^{-1}(v_i)$ would already have a cycle, which we have assumed is not the case. Therefore, the cycle must include an edge connecting nodes $w_i$ and $w_{i+1}$ where $dep(w_i) \neq dep(w_{i+1})$. Note that $w_i, w_{i+1} \notin \{v_1, \ldots, v_n\}$ and that $dep(w_i), dep(w_{i+1}) \in \{v_1, \ldots, v_n\}$. Such an edge cannot exist as it fails to satisfy the second property in the definition of ego-rank.

2. We assume $dep$ satisfies property 1. Let $dep(w) = v$ and suppose that property 2 fails, i.e., there is a node $w$ with $dep(w) = v$ such that no node $w'$ reachable from $w$ in the subgraph induced by $dep^{-1}(v)$ is adjacent to $v$. Let $dep'$ be identical to $dep$ except that, for all $w'$ reachable from $w$ in the subgraph induced by $dep^{-1}(v)$, we set $dep'(w') := dep(v)$. Note that, by property 1, $v \notin deps(v)$ and hence the net effect of this change is that $deps'(w') = deps(w') \setminus \{v\}$. We claim that $deps'$ still satisfies all requirements from the definition of ego-rank. Indeed:

- $dep'(u)$ is still $\bot$.

- Let $(w_1, w_2) \in E$ be an edge. Then one of the following conditions holds:

  (a) $dep(w_1) = dep(w_2)$. Then the same holds for $dep'$ (note that $w_1$ and $w_2$ belong to the same connected component of $dep^{-1}(dep(w_1))$).

  (b) $w_1 \in deps(w_2)$ or vice versa. It is easy to see that, in this case, the same still holds for $deps'$.

- Finally, we must show that, for each $x \in V \cup \{\bot\}$, the subgraph induced by $dep'^{-1}(x)$ is acyclic. It suffices to consider the case where $x = dep(v)$, because, for all other $x$ we have that $dep'^{-1}(x) \subseteq dep^{-1}(x)$. Therefore, let $x = dep(v)$ and suppose for the sake of a contradiction that $dep'^{-1}(x)$ contains a cycle. Since $dep^{-1}(x)$ and $dep^{-1}(v)$ were both acyclic, this cycle must include an edge $(w_1, w_2)$ such that $dep(w_1) = v$ and $dep(w_2) = x$. It must then be the case that $w_2 \in deps(w_1)$, and, indeed, it must be the case that $w_2 = v$, a contradiction since the connected component of $w_1$ in $dep^{-1}(v)$ was not supposed to be connected to $v$.

By repeating this operation, we obtain a dep-function satisfying property 2. $\qquad\square$

**Proposition D.6.** *The rooted $n \times 2$-grid (with $n \geq 1$) as depicted in Figure 2 has ego-rank $n - 1$.*

*Proof.* For the $n - 1$ upper bound, a witnessing $dep$-function is already depicted in Figure 2 for the special case of $n = 5$, and it can be modified in the obvious way for the general case (note that there are also other choices for the $dep$ function that yield the same ego rank). In what follows, we prove the $n - 1$ lower bound.

Let $dep : V \to V \cup \{\bot\}$ be any function satisfying the requirements in the definition of ego-rank. We may assume that it also satisfies the properties described in Lemma D.5.

Recall that $u_1$ is the root of the rooted graph. We show there is a sequence

$$\langle \pi_1, \pi_2, \ldots, \pi_n \rangle$$

where $\pi_1 = u_1$, and for $i > 1$ the following holds:

1. $\pi_i \in \{u_i, v_i\}$

2. Either (i) $dep(\pi_i) = \pi_{i-1}$ or (ii) $dep(\pi_i) = \pi'_{i-1}$ and $dep^2(\pi_i) = \pi_{i-1}$

where $\pi'_{i-1} = \begin{cases} u_{i-1} & \text{if } \pi_{i-1} = v_{i-1} \\ v_{i-1} & \text{if } \pi_{i-1} = u_{i-1} \end{cases}$

We apply induction over $i > 1$. It follows from the induction hypothesis that $deps(\pi_{i-1})$ doesn't contain any $u_j, v_j$ with $j \geq i - 1$. Suppose w.l.o.g. that $\pi_{i-1} = u_{i-1}$. Note firstly that either $u_{i-1} \in deps(v_{i-1})$ or $u_{i-1} \in deps(u_i)$. For, if this would not hold then, by definition of $dep$, $dep(u_{i-1}) = dep(v_{i-1}) = dep(u_i)$, which by well-foundedness leaves no possibility for $dep(v_i)$.

Since every connected component of $dep^{-1}(u_{i-1})$ is connected to $u_{i-1}$ it follows that $u_{i-1} = dep(u_i)$ or $u_{i-1} = dep(v_{i-1})$. In the first case we let $\pi_i = u_i$.

Suppose then that $u_{i-1} = dep(v_{i-1})$, then either $u_{i-1} = dep(v_i)$ or $v_{i-1} \in deps(v_i)$. In the second case, by well-foundedness and the connectedness of $dep^{-1}(v_{i-1})$ to $v_{i-1}$ it follows that $v_{i-1} = dep(v_i)$. In both cases we let $\pi_i = v_i$, completing the induction.

Since $|deps(\pi_n)| \geq n - 1$ the lower bound follows. $\qquad\square$

**Lemma D.7.** *Let $(F, u)$ be a rooted graphs and $N > 0$. Then there exists a number $M$ such that, for all pointed graphs $(G, v)$ of degree at most $N$, $hom((F, u), (G, v)) \leq M$.*

*Proof.* Let $r$ be the maximal distance from $u$ to any other node of $F$. The radius-$r$ neighborhood of $v$ in $G$ contains at most $(N + 1)^r$ nodes. It follows that there can be at most $M = ((N + 1)^r)^n$ homomorphisms from $(F, u)$ to $(G, v)$, where $n$ is the number of nodes of $F$. $\qquad\square$

**Lemma D.8.** ReLU-*FFNNs can multiply small natural numbers. That is, for each $N > 0$ and $k > 0$, there is a* ReLU-*FFNN with input dimension $k$ and output dimension $1$ that, on input $(x_1, \ldots, x_k)$ with $0 \leq x_i \leq N$, outputs $\Pi_i x_i$.*

*Proof.* This is well-known (and holds not only for ReLU but also for other common non-linear activation functions). For the sake of completeness we sketch a proof. First, for a fixed $m$, we can test with a ReLU-FFNN, for a given natural number $x$, whether $x \geq m$. Indeed, $\text{ReLU}(1 - \text{ReLU}(m - x_i))$ is 1 if this holds and 0 otherwise. Furthermore, the Boolean operators of conjunction and negation can be implemented by ReLU-FFNNs as well (cf. the proof of Proposition 2.3). It follows that the function

$$f_{(m_1, \ldots, m_k)}(x_1, \ldots, x_k) = \begin{cases} 1 & \text{if } x_i = m_i \text{ for all } i \leq k \\ 0 & \text{otherwise} \end{cases}$$

can also be implemented by a ReLU-FFNN. Using this, we can represent the product $\Pi_i x_i$ by the linear expression

$$\sum_{0 \leq m_1, \ldots, m_k \leq N} \left( \Pi_i m_i \cdot f_{(m_1, \ldots, m_k)}(x_1, \ldots, x_k) \right)$$

Note that the $\Pi_i m_i$ factors are viewed as constant integer coefficient here. $\qquad\square$

**Proposition 5.7.** *For all rooted graphs $(G, v)$ with $G = (V, E, \text{lab})$,*

1. *tree-width$(G) - 1 \leq$ ego-rank$(G, v) \leq |V|$.*

2. *ego-rank$(G, v) = 0$ if and only if $G$ is acyclic.*

3. *ego-rank$(G, v) = 1$ whenever $(G, v)$ is c-acyclic.*

*Proof.* For the first part of the first claim, let $dep$ be a function witnessing that $(G, v)$ has ego-rank $k$. We define a tree decomposition as follows:

- The nodes of the tree decomposition are (i) all nodes $w$ of the graph $G$, and (ii) all edges $(w, v)$ of the graph $G$ satisfying $dep(w) = dep(v)$. The bag associated to each $w$ is $\{w\} \cup deps(w)$ and the bag associated to each edge $(w, v)$ is $\{w, v\} \cup deps(w)$.

- The edges of the tree decomposition are pairs where one node is a node of $G$ and the other is an edge of $G$ in which the node participates.

Note that, in this way, every edge of $G$ is indeed contained in a bag. Furthermore, the third condition in the definition of dependency functions guarantees that this tree decomposition is indeed a tree.

Consider any path in the tree decompositions of the form

$$w_1 \, (w_1, w_2) \, w_2 \, (w_2, w_3) \, \ldots (w_{n-1}, w_n) \, w_n$$

where $dep(w_i) = dep(w_j)$ for all $i < j < n$, and hence $w_i \neq w_j$ for all $i < j < n$ (because the subgraph induced by $dep^{-1}(dep(w_i))$ is acyclic). Suppose, now, that some graph vertex $x$ belongs to the bag of $w_1$ as well as to the bag of $w_n$. In this case, $x$ must belong to $deps(w_1)$ and to $deps(w_n)$ and hence it belongs to the bag of each node on the path. A similar argument applies for paths that start or end with an edge. This shows that the constructed tree decomposition is indeed a valid tree-decomposition. Moreover, the maximal bag size is $k + 2$. Therefore, the tree-width of $G$ is at most $k + 1$.

For the second part of the first claim, it suffices to choose an arbitrary enumeration $v_1, v_2, \ldots v_n$ of the nodes of $G$, where $v_1 = v$, and set $dep(v_1) = \bot$ and $dep(v_{i+1}) = v_i$.

The second claim follows immediately from the definition of ego-rank.

For the third claim, it suffices to take $dep(v) = \bot$ and $dep(v') = v$ for all $v' \neq v$. $\qquad\square$

The next theorem is not stated in the body of the paper, but it's a special case of Theorem 5.9(1) below, and it serves as a warming up towards the proof of Theorem 5.9.

**Theorem D.9.** *Let $\mathfrak{F}$ be any finite set of acyclic rooted graphs. There is a $(|P|, |P| + |\mathfrak{F}|)$-GNN $\mathcal{A}$ such that, for all pointed graphs $(G, v)$, $\mathrm{run}_{\mathcal{A}}(G)(v) = \mathrm{emb}_G(v) \oplus \hom(\mathfrak{F}, (G, v))$. The GNN in question uses multiplication in the combination function.*

*Proof.* In what follows, we will refer to acyclic rooted graphs $(F, u)$ also as *trees*, where we think of $u$ as the root of the tree. By an *immediate subtree* of a tree $(F, u)$ we will mean a rooted graph $(F', u')$ where $u'$ is a neighbor of $u$ and where $F'$ is the induced subgraph of $F$ consisting of all nodes whose shortest path to $u$ contains $u'$. We will write $(F, u) \Rightarrow (F', u')$ to indicate that $(F', u')$ is an immediate subtree of $(F, u)$. By the *depth* of a tree $(F, u)$ we will mean the maximum, over all nodes $u'$ of $F$, of the distance from $u$ to $u'$. Note that $(F, u)$ has depth zero if and only if it has no immediate subtrees, and that the depth of an immediate subtree of $(F, u)$ is always strictly smaller than the depth of $(F, u)$.

We may assume without loss of generality that $\mathfrak{F}$ is closed under taking immediate subtrees. The general case then follows by adding one additional layer on top that projects the resulting embedding vectors to a subset of $\mathfrak{F}$.

Let $|\mathfrak{F}| = \{(F_1, u_1), \ldots, (F_k, u_k)\}$ and let $L$ be the maximal depth of a rooted graph in $\mathfrak{F}$. Let

$$\mathcal{A} = ((\mathrm{COM}_i)_{i=1\ldots L}, (\mathrm{AGG}_i)_{i=1\ldots L})$$

where

- $\mathrm{COM}_1 : \mathbb{R}^{2|P|} \to \mathbb{R}^{|P|+k}$ given by $\mathrm{COM}_1(x_1, \ldots, x_{|P|}, z_1, \ldots, z_{|P|}) = (y_1, \ldots, y_{|P|+k})$ with $y_i = x_i$ for $i \leq |P|$, and with

$$y_{|P|+i} = \begin{cases} 1 & \text{if } F_i \text{ has depth 0 and } x_j = 1 \text{ for each } p_j \in \mathrm{lab}^{F_i}(u_i) \\ 0 & \text{otherwise} \end{cases}$$

- For $i > 1$, $\mathrm{COM}_i : \mathbb{R}^{2(|P|+k)} \to \mathbb{R}^{|P|+k}$ given by $\mathrm{COM}_i(x_1, \ldots, x_{|P|+k}, z_1, \ldots, z_{|P|+k}) = (y_1, \ldots, y_{|P|+k})$ with $y_i = x_i$ for $i \leq |P|$, and with

$$y_{|P|+i} = \begin{cases} \displaystyle\prod_{F_j \text{ with } F_i \Rightarrow F_j} (z_{|P|+j}) & \text{if } x_\ell = 1 \text{ for each } p_\ell \in \mathrm{lab}^{F_i}(u_i) \\ 0 & \text{otherwise} \end{cases}$$

- Each $\mathrm{AGG}_i$ is (pointwise) sum.

It is not difficult to see that $\text{COM}_i$ can be implemented by a FFNN using ReLU and multiplication.

It follows from the above construction, and by induction on $d$, that, for all $d \geq 0$ and for all $(F_j, u_j) \in \mathfrak{F}$ of depth $d$, $\text{emb}_G^i(v)(|P| + j) = \text{hom}((F_j, u_j), (G, v))$ for $i > d$. Furthermore, it is immediately clear from the construction that $\text{emb}_G^i(v)(j) = \text{emb}_G(v)(j)$ for all $j \leq |P|$. $\qquad \square$

**Theorem 5.9.** *Let $\mathfrak{F}$ be any finite set of rooted graphs, let $d = \max\{ego\text{-}rank(F, u) \mid (F, u) \in \mathfrak{F}\}$.*

1. *Then there is a $(|P|, |P| + |\mathfrak{F}|)$-HE-GNN $\mathcal{A}$ of nesting depth $d$ such that, for all pointed graphs $(G, v)$, $\text{run}_\mathcal{A}(G)(v) = \text{emb}_G(v) \oplus \text{hom}(\mathfrak{F}, (G, v))$. The HE-GNN uses multiplication in the combination functions.*

2. *For each $N > 0$, there is a $(|P|, |P| + |\mathfrak{F}|)$-HE-GNN $\mathcal{A}$ of nesting depth $d$ such that, for pointed graphs $(G, v)$ of degree at most $N$, $\text{run}_\mathcal{A}(G)(v) = \text{emb}_G(v) \oplus \text{hom}(\mathfrak{F}, (G, v))$. The HE-GNN uses only Sum as aggregation and ReLu-FFNNs as combination functions.*

*Proof.* We prove the first statement. The proof of the second statement is identical, except that we can replace the use of multiplication by a ReLU-FFNN due to the fact that the numbers being multiplied are bounded by a constant (cf. Lemma D.7 and Lemma D.8).

We may assume that $\mathfrak{F}$ consists of a single rooted graph $(F, u)$. Let $dep$ be the dependency function witnessing the fact that $(F, u)$ has ego-rank $d$. By Lemma D.5, we may assume that $dep$ is well-founded, and that for each node $w$, if $dep^{-1}(w)$ is non-empty, then each connected component of the subgraph induced by $dep^{-1}(w)$ contains a neighbor (in the original graph $F$) of $w$.

By the *dependency depth* of a node $w$ of $F$, we will mean the largest number $\ell$ for which it is the case that $w = dep^\ell(w')$ for some $w'$. Note that this is a finite number.

For a node $w$, we will denote by $F_w$ the subgraph of $F$ induced by the set of nodes $deps^{-1}(w) \cup \{w\} \cup deps(w)$, where $deps^{-1}(w)$ is the set of nodes $w'$ for which $w \in deps(w)$.

We will now prove the following claim:

(*) For each node $w$ of dependency depth $\ell \geq 1$ with $deps(w) = \{w_1, \ldots, w_k\}$, there is a $(|P| + k + 1, 1)$-HE-GNN $\mathcal{A}_w$ of nesting depth $\ell - 1$ such that for all graphs $G$ with vertices $V_G$ and maps $h : \{w\} \cup deps(w) \rightarrow V_G$,

$$\text{run}_\mathcal{A}(G, \text{emb}_G^{+h})(h(w)) = \text{hom}_h(F_w, G)$$

where $\text{emb}_G^{+h} = \{v' : \text{emb}_G(v') \oplus \langle \delta_{v'h(w_1)}, \ldots, \delta_{v'h(w_k)}, \delta_{v'h(w)} \rangle \mid v' \in V_G\}$.

Recall that $\text{hom}_h(F_w, G)$ denotes the number of homomorphisms from $F_w$ to $G$ extending the partial function $h$. Also, note that the embedding vectors of $\text{emb}_G^{+h}$, by construction, include features that uniquely mark the $h$-image of each $w' \in deps(w)$ as well as $w$.

We will first prove (*) by induction on the dependency depth $\ell \geq 1$ of $w$, and then show that it implies the main statement of our theorem.

The subgraph $F_w$ can be decomposed as described in Figure 7. By Lemma D.5, each connected component of the subgraph induced by $dep^{-1}(w)$ of $F$ includes at least one neighbor of $w$. Let $(F', w)$ be the rooted tree that consists of the subgraph induced by $\{w\} \cup dep^{-1}(w)$, after removing, for each connected component of $dep^{-1}(w)$, all but one (arbitrarily chosen) connecting edge to $w$. Note that, as explained in the caption of Figure 7, there may be more than one edge between $w$ and a given connected component of $dep^{-1}(w)$, but we only keep one in order to ensure that $F'$ is a tree rooted at $w$. We refer to the edges connecting $w$ to nodes in $dep^{-1}(w)$ that we did not add, as well as edges connecting nodes in $deps(w)$ to nodes in $dep^{-1}(w)$ as "back-edges".

Now, by construction, for every function $h : \{w\} \cup deps(w) \rightarrow V_G$, we have that

$$\text{hom}_h(F_w, G) = \sum_{\text{homomorphisms } f \,:\, (F', w) \,\rightarrow\, (G, h(w))} \quad \prod_{w_i \in dep^{-1}(w)} \text{hom}_{h \cup \{(w_i, f(w_i))\}}(F_{w_i}, G)$$

Note that the fact that we omitted the back-edges from $F'$ does not impact the above equation. Indeed, if a function $f : (F', w) \rightarrow (G, h(w))$ fails to preserve a back-edge it will simply not extend to a homomorphism from $F_{w_i}$ to $G$, and hence will not contribute to the above sum.

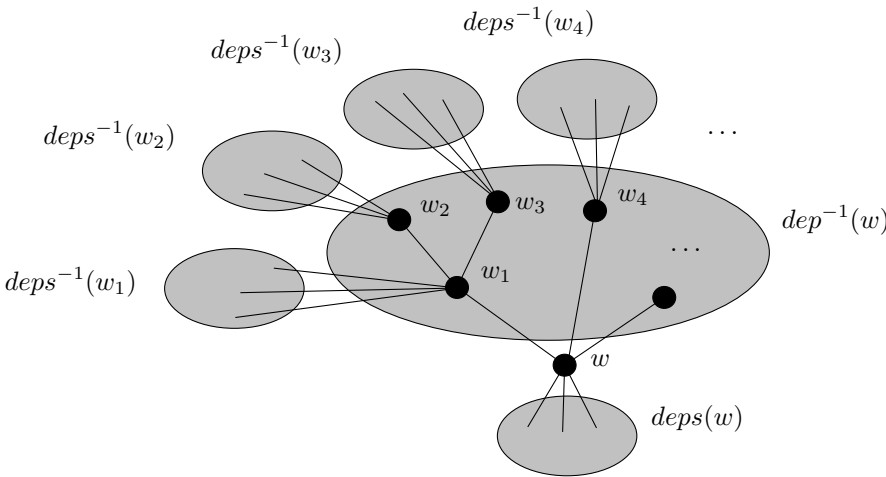

Figure 7: Decomposition of the subgraph $F_w$ induced by $deps(w) \cup \{w\} \cup deps^{-1}(w)$ when $w$ has dependency depth greater than 1. The gray circles are disjoint. In addition to the edges drawn in the picture, there may also be additional edges connecting nodes in $deps^{-1}(w)$ to nodes in $\{w\} \cup deps(w)$. We omitted such edges from the drawing in order to not clutter the picture.

Now if $\ell = 1$, $\hom_{h \cup \{w_i, f(w_i)\}}(F_{w_i}, G)$ is either 1 or 0 since every vertex of $F_{w_i}$ is in the domain of $h \cup \{(w_i, f(w_i))\}$. Since all nodes in the range of this function are uniquely marked, this can be computed by a $(|P| + k + 1, 1)$- GNN $\mathcal{C}_{w_i}$, i.e. a HE-GNN of nesting depth 0. More precisely, for each $v' \in G$:

$$\text{run}_{\mathcal{C}_{w_i}}(G, \text{emb}_G^{+h \cup \{(w_i, v')\}})(v') = hom_{h \cup \{(w_i, v')\}}(F_{w_i}, G)$$

These can be combined using Lemma B.3 into a single GNN $\mathcal{C}$ computing their concatenation. We now add layers, exactly as in the proof of Theorem D.9 that run through the tree $(F', w)$ and aggregate the respective counts according to the above equation to compute $\hom_h(F_w, G)$.

If $l > 1$ we can apply the induction hypothesis (*) to obtain a $(|P| + k + 2, 1)$-HE-GNN $\mathcal{B}_{w_i}$ of nesting depth at most $\ell - 2$ for each $w_i \in dep^{-1}(w)$, calculating the corresponding factor in the above equation. Now for each $v' \in G$:

$$\text{run}_{\mathcal{B}_{w_i}}(G, \text{emb}_G^{+h \cup \{(w_i, v')\}})(v') = hom_{h \cup \{(w_i, v')\}}(F_{w_i}, G)$$

These can be combined using Lemma B.4 into a single HE-GNN $(\mathcal{B}, \mathcal{C})$ of nesting depth $\ell - 2$ computing their concatenation. Now similarly to the case for $\ell = 1$, we add layers to $\mathcal{C}$ that run through the tree $(F', w)$ and aggregate the respective counts according to the above equation and call the result $\mathcal{C}'$. Let $\mathcal{I}$ be a trivial GNN. Then $\mathcal{A} = ((\mathcal{B}, \mathcal{C}'), \mathcal{I})$ is a $(|P| + k + 1, 1)$-HE-GNN of nesting depth $\ell - 1$ that computes $\hom_h(F_w, G)$, concluding the proof by induction of (*).

Finally, we must show that (*) implies the main statement of our theorem. Let $(F', u)$ be the induced subgraph of $F$ with nodes in $dep^{-1}(\bot)$. Then again:

$$\hom((F, u), (G, v)) = \sum_{\text{homomorphisms } f : (F', u) \to (G, v)} \prod_{w_i \in F'} \hom_{\{(w_i, f(w_i))\}}(F_{w_i}, G)$$

The argument now is very similar as the one we used in the inductive step. Since each $w_i \in F'$ has dependency depth at most $d$, by (*) and theorem D.9 we obtain for each $w_i \in F'$, a $(|P| + 1, 1)$-HE-GNN $\mathcal{B}_{w_i}$ of nesting depth $d - 1$ computing the corresponding factor in the above equation. These can be combined using Lemma B.4 into a single $(|P| + 1, |F'|)$-HE-GNN $(\mathcal{B}, \mathcal{C})$ of nesting depth $d - 1$ computing their concatenation. We again add layers to $\mathcal{C}$ as in the proof of Theorem D.9 that run through the tree $(F', u)$ and aggregate the respective counts according to the above equation, and call this GNN $\mathcal{C}'$. Then for trivial GNN $\mathcal{I}$, It follows that $\mathcal{A} = ((\mathcal{B}, \mathcal{C}'), \mathcal{I})$ is a $(|P|, 1)$-HE-GNN of nesting depth $d$ that computes $\hom((F, u), (G, v))$. $\qquad\square$

### 5.3. Higher order GNNs

**Theorem 5.11.** *For $d \geq 0$, $\rho((d + 1)\text{-WL}) = \rho((d + 2)\text{-GNN }) \subseteq \rho(\text{HE-GNN-}d)$.*

*Proof.* Let $\mathcal{A}$ be a HE-GNN with depth $d$ such that $\mathsf{cls}_{\mathcal{A}}(G, v) \neq \mathsf{cls}_{\mathcal{A}}(G', v')$. By theorem 3.4 there exists a GML($\downarrow^d$)-sentence $\phi$ such that $\mathsf{cls}_{\phi}(G, v) \neq \mathsf{cls}_{\phi}(G', v')$.

$\phi$ is equivalent to a sentence that uses at most $d$ variables $x_i$. Hence there exists an equivalent formula $\psi$ in the $d + 2$-variable fragment of first-order logic with counting quantifiers (cf. Table 2). By theorem 5.10 there then exists a $(d + 2)$-GNN $\mathcal{B}$ such that $\mathsf{cls}_{\mathcal{B}}(G, v) \neq \mathsf{cls}_{\mathcal{B}}(G', v')$. $\qquad\square$

We apply a lemma by Morris et al. [37], which was slightly adjusted by Qian et al. [42].

**Definition D.10.** *For a graph* $G = (V, E, \text{lab})$, $S \subseteq V$ *forms a* distance 2 clique *if any two nodes in $S$ are connected by a path of length* 2. *We say $S$ is* colorful *if for all $v, v' \in S$, $\text{lab}(v) = \text{lab}(v')$ iff $v = v'$.*

**Lemma D.11.** *For $d \in \mathbb{N}$ there exist graphs $G_d, H_d$ such that:*

1. $G_d \equiv_{(d-1)\text{-WL}} H_d$

2. *There exists a colorful distance 2 clique of size $d + 1$ in $G_d$.*

3. *There does not exist a colorful distance 2 clique of size $d + 1$ in $H_d$.*

Table 2: Translation from a GML($\downarrow$)-formula $\phi$ containing variables $z_1, \ldots, z_k$ to a $\mathsf{C}^{k+2}$-formula $tr_x(\phi)$ containing variables $x, y, z_1, \ldots, z_k$.

| | | | | | |
|---|---|---|---|---|---|
| $tr_x(p_i)$ | $=$ | $P_i(x)$ | $tr_y(p_i)$ | $=$ | $P_i(x)$ |
| $tr_x(z_i)$ | $=$ | $x = z_i$ | $tr_y(z_i)$ | $=$ | $y = z_i$ |
| $tr_x(\phi \wedge \psi)$ | $=$ | $tr_x(\phi) \wedge tr_x(\psi)$ | $tr_y(\phi \wedge \psi)$ | $=$ | $tr_y(\phi) \wedge tr_y(\psi)$ |
| $tr_x(\neg \phi)$ | $=$ | $\neg tr_x(\phi)$ | $tr_y(\neg \phi)$ | $=$ | $\neg tr_y(\phi)$ |
| $tr_x(\Diamond^{\geq n}\phi)$ | $=$ | $\exists^{\geq n} y(Rxy \wedge tr_y(\phi))$ | $tr_y(\Diamond^{\geq n}\phi)$ | $=$ | $\exists^{\geq n} x(Ryx \wedge tr_x(\phi))$ |
| $tr_x(\downarrow z_i.\phi)$ | $=$ | $\exists z_i(z_i = x \wedge tr_x(\phi))$ | $tr_y(\downarrow z_i.\phi)$ | $=$ | $\exists z_i(z_i = y \wedge tr_y(\phi))$ |

**Theorem 5.12.** *For $d \geq 0$, HES-GNN-($d$,3) can distinguish pointed graphs that cannot be distinguished by $d$-WL, or equivalently, by a $(d + 1)$-GNN.*

*Proof.* We show there exist formula $\phi \in \text{GML}(\downarrow^d_{W^3})$ and node $v$ in $G_{d+1}$ such that $G_{d+1}, v \models \phi$ but $H_{d+1}, w \not\models \phi$ for all $w$ in $H_{d+1}$.

Let $v$ be in a distance 2 colorful clique $S$ of size $d + 2$, and let $\alpha_1 \ldots \alpha_{d+2}$ be conjunctions of literals matching the distinct labelings of nodes in $S$. Thus $v, v'$ only satisfy the same $\alpha_i$ if $\text{lab}(v) = \text{lab}(v')$. Further let $G_{d+1}, v \models \alpha_1$.

We let:
$$\phi = \downarrow x_1.W^3(\Diamond\Diamond \downarrow x_2.W^3(\Diamond\Diamond \downarrow x_3.W^3(\ldots, \downarrow x_d.W^3(\xi \wedge \psi)\ldots)))$$

$\xi$ ensures that all $x_i$ have distinct values matching labeling as specified by $\alpha_1, \ldots \alpha_d$, and form a colorful distance 2 clique of size $d$.
$$\xi = \bigwedge_{1 \leq i \leq d} (@_{x_i}(\alpha_i \wedge \bigwedge_{1 \leq j \leq d, j \neq i} (\Diamond\Diamond x_j)))$$

$\psi$ ensures that there are two more connected vertices with labelings matching $\alpha_{d+1}, \alpha_{d+2}$ that have edges to all the $x_i$:
$$\psi = \Diamond\Diamond(\alpha_{d+1} \wedge \bigwedge_{1 \leq i \leq d} (\Diamond\Diamond x_i)) \wedge \Diamond\Diamond(\alpha_{d+2} \wedge \bigwedge_{1 \leq i \leq d} (\Diamond\Diamond x_i))$$

Now $G_{d+1}, v \models \phi$ and any node in $H_{d+1}$ satisfying $\phi$ would be in a distance 2 colorful clique. Hence for all $v'$ in $H_{d+1}$:
$$G_{d+1}, v \not\equiv_{\text{GML}(\downarrow^d_{W^2})} H_{d+1}, v'$$
We apply the logical characterization of HES-GNN (Theorem 4.1), the WL characterization of higher order GNNs and the fact that $G_{d+1} \equiv_{(d-1)\text{-WL}} H_{d+1}$ to obtain the theorem:
$$\rho((d+1)\text{-GNN}) = \rho(d\text{-WL}) \not\subseteq \rho(\text{GML}(\downarrow^d_{W^3})) = \rho(\text{HES-GNN-}(d,3))$$

$\qquad\square$

