# OpenReview forum: "Logical Expressiveness of Graph Neural Networks with Hierarchical Node Individualization"
_NeurIPS.cc/2025/Conference — NeurIPS 2025 poster_

### Official Review · Reviewer_Ru4p · 2025-06-23

**Clarity:** 3
**Significance:** 3
**Originality:** 3
**Rating:** 4
**Confidence:** 4

**Summary:**

This paper introduces Hierarchical Ego Graph Neural Networks (HE-GNNs), a novel class of GNNs that enhance expressive power through a mechanism inspired by the Individualization-Refinement (IR) paradigm from graph isomorphism testing. HE-GNNs iteratively individualize nodes and apply message-passing, resulting in a hierarchy of models where increasing the nesting depth increases expressive power. The authors provide a tight logical characterization of HE-GNNs using graded hybrid logic (GML($\downarrow$)) and its subgraph-restricted variant (GML($\downarrow$,W)). They show that HE-GNNs match or surpass existing expressive models such as higher-order GNNs, subgraph-GNNs, and homomorphism-based GNNs. The theoretical results are backed by empirical experiments showing HE-GNNs can distinguish graphs that standard GNNs fail on, including strongly regular graphs and graphs in datasets like ZINC.

**Questions:**

1. Have you considered evaluating HE-GNNs on synthetic isomorphism benchmarks like SR25, CSL, or EXP to better demonstrate expressiveness in challenging graph distinction tasks?
2. Can you provide empirical evidence on how performance and computational cost scale with increased nesting depth $d$ and radius $r$?
3. Theorem 4.1: Could you elaborate on how the quantifier radius in GML($\downarrow$, W) corresponds exactly to the radius-$r$ subgraphs in HES-GNNs? Are there cases where the logical quantification covers more nodes than the subgraph accessible by radius-$r$ message passing?
4. Theorem 4.3: Is the strict increase in expressiveness dependent on the existence of graphs where node distinctions require radius > $r$ neighborhoods? Could you provide such a minimal pair of graphs or clarify how often such cases occur in real-world datasets?
5. While you show that 2-WL and Nested GNNs are subsumed by HES-GNNs under certain conditions, could you clarify whether this inclusion is strict in all cases or only under specific configurations of depth and radius?

**Ethical Concerns:**

["NO or VERY MINOR ethics concerns only"]

**Final Justification:**

I appreciate the authors' thorough and well-reasoned responses. I am convinced by most of the clarifications provided in the rebuttal. Additionally, the promised ablation studies, runtime analysis, and the inclusion of synthetic isomorphism benchmarks are positive steps toward strengthening the empirical side of the paper. Thus, I will raise my score accordingly.

**Limitations:**

yes

**Quality:**

3

**Strengths And Weaknesses:**

**Strengths**
1. HE-GNNs is a compelling and novel approach that generalizes and unifies various expressive GNN architectures under the lens of logic. The integration of hybrid logic to describe the expressive capabilities of GNNs is particularly innovative.
2. The paper is rigorous in its formalism. The equivalence theorems connecting HE-GNNs with graded hybrid logic (Theorems 3.3, 4.1) are carefully proved and grounded in prior foundational results (e.g., Barcelo et al., Morris et al.). The use of logical tools for characterization, such as nesting-depth correspondence, is both insightful and technically strong.
3. This work is foundational, it provides a formal framework to measure and compare the expressive power of advanced GNN architectures. The logical perspective opens new research directions for designing more powerful and interpretable models. Also, HE-GNNs offer a scalable alternative to higher-order GNNs with bounded memory overhead.

**Weaknesses**
1. The experimental validation is relatively limited in breadth. While the ZINC and SRG evaluations support the theory, inclusion of synthetic benchmarks like CSL, EXP, and SR25 could offer a clearer picture of the model's capabilities in distinguishing structurally similar yet non-isomorphic graphs.
2. The empirical evaluation lacks diversity in real-world benchmarks. The model is only evaluated on a single real-world dataset (ZINC), with no experiments on large-scale and widely used benchmarks such as the OGB datasets. This limits the ability to assess the practical effectiveness, scalability, and generalizability of HE-GNNs in more complex and varied real-world scenarios.
3. Although HE-GNNs are presented as more efficient than higher-order GNNs, no detailed comparison of runtime, memory, or practical training complexity is provided.
4. The impact of depth $d$ and radius $r$ in HES-GNNs is not fully analyzed in experiments (missing ablation study). A study on how performance and runtime vary with $d$ and $r$ would strengthen the claims.

---

> ### Author Rebuttal · Authors · 2025-07-30
>
> We thank the reviewer for their constructive feedback, and have revised our manuscript to address the points raised. We appreciate that the reviewer considers the paper to be foundational, allowing for comparison of existing GNN architectures and opening up new research directions, since this is precisely what we aim to achieve with this work.
>
> **Experimental validation**
>
> We have focused on formalizing and presenting the capacity of HE-GNN to express existing architectures, and recognize that further experimental validation would strengthen the paper. We vouch to:
>
> - Attempt comparison on an additional synthetic dataset like CSL, addressing question 1.
>
> - Do our best to compare performance on ZINC with other modern node-marking based subgraph GNNs.
>
> - Provide an ablation study with results for varying $d,r$, to show how HES-GNNs transition from standard message-passing networks to more expressive architectures in terms of both accuracy and running time, addressing question 2.
>
> In addition, we have made the complexity analysis and comparison with higher-order GNNs more precise.
>
> **Further responses to questions**
>
> **Response to question 3**
>
> In question 3 the reviewer asks how the quantifier radius $r$ in GML$(\downarrow_W)$ corresponds to the subgraph radius in HE-GNN. We can confirm that both notions refer to the same subgraph: the graph induced by all nodes reachable in $r$ steps. However, if a specific HE-GNN has fewer than $r$ message-passing iterations, not all nodes in this subgraph are included in the computation. This corresponds to evaluating a GML$(\downarrow_W)$ with bounded $\diamond$ depth.
>
> **Response to question 4**
>
> In question 4, the reviewer asks whether we can provide a pair of graphs that require a minimal neighborhood radius to be separated by HES-GNN($d$,$r$), making the strictness of theorem 4.3 concrete.
>
> Such a pair of graphs is provided in the proof of theorem 5.11. For each $d$, there exists a GML$(\downarrow^d_{W^3})$ formula that can recognize a ‘colorful distance 2 clique’ of size $d+2$. No such formula exists with smaller radius, since a radius $2$ subgraph cannot contain this structure. Further, no such formula exists with smaller depth, as established by theorem 5.10.
>
> **Response to question 5**
>
> In question 5 the reviewer asks if HES-GNN strictly subsumes 2-WL either in general, or under specific $d$ and $r$.
>
>  Theorem 5.11 shows that HES-GNN with depth $d$ can separate graphs that $d$-WL cannot separate. This result holds for any $r \geq 3$, and we are not aware of a construction with smaller subgraph radius. We have not shown however that HES-GNN with any $d$ or $r$ subsumes the separating power of 2-WL.
>
> Note that we use WL for what some authors name Folklore-WL, as opposed to Oblivious-WL (see e.g. Morris et al. [1]). If the reviewer refers to 2-Oblivious-WL, then theorem 5.11 indeed show that HES-GNN($d$,$r$) strictly subsumes this test for $d \geq 1$, $r \geq 3$.
>
> [1] Morris, C., Lipman, Y., Maron, H., Rieck, B., Kriege, N. M., Grohe, M. & Borgwardt, K. (2023). Weisfeiler and leman go machine learning: The story so far. Journal of Machine Learning Research, 24(333), 1-59.

---

> > ### Comment · Reviewer_Ru4p · 2025-08-04
> > **Follow-Up to Rebuttal**
> >
> > Thank you for your detailed and thoughtful rebuttal. I appreciate the clarification on how the quantifier radius in GML($\downarrow$ W) maps to the message-passing radius in HES-GNNs, and the concrete examples supporting Theorems 4.3 and 5.11. The promised ablation study, runtime analysis and the inclusion of CSL are positive steps toward strengthening the empirical side of the paper.
> >
> > To further clarify the theoretical positioning and limitations of the proposed model, I have the following follow-up questions:
> >
> > 1. **Subsumption of 2-WL**: While the authors claim that HES-GNNs can strictly separate some 2-WL-indistinguishable graphs, it remains unclear whether HE-GNNs strictly subsume 2-WL on all graph classes or only separate a strict subset depending on depth and radius parameters. Could you clarify the scope and limitations of this separation?
> >
> > 2. **Position in the WL Hierarchy**: Where does HE-GNNs lie in the broader WL hierarchy? Is there a known upper bound on their expressiveness, for instance, do they fall strictly below 3-WL or match any known refinement procedure?
> >
> > 3. **Role of Node Individualization vs. Color Refinement**: The model relies heavily on hierarchical node individualization. Could similar expressiveness be achieved through color refinement alone? Can you formally characterize the added power (if any) that node marking provides over classical refinement methods?
> >
> > 4. **Logical vs. Learned Approximation Gap**: While GML($\downarrow$ W) characterizes the logical expressiveness of HE-GNNs, is there a formal or empirical gap between this and the behavior of finite-depth, parameterized HE-GNNs trained via gradient-based optimization? Are there graph pairs that are theoretically separable under GML logic, but not practically distinguishable by trained HE-GNNs due to optimization limitations?

---

> > > ### Author Response · Authors · 2025-08-04
> > >
> > > Thank you for the addition
> > >
> > > 1. **Subsumption of 2-WL**
> > >
> > > HE-GNN does not subsume $2$-WL (unless we mean $2$-oblivious-WL, which has the separating power of standard GNNs).
> > >
> > > 2. **Position in the WL hierarchy**
> > >
> > > We have shown HE-GNN with depth $d$ to be contained in $d+1$ WL, but not contained in $d$ WL for each $d \geq 1$. This also holds for HES-GNN with depth $d$ and radius of at least $3$. It remains open whether HE-GNN with depth $d$ subsumes any $d'$-WL with $1< d' \leq d$, or whether HE-GNN is incomparable with all such $d'$-WL.
> > >
> > > 3. **Node Individualization vs Color Refinement**
> > >
> > > This is an interesting question that we have not considered, but we found an answer via GML($\downarrow_W$). If we remove individualization, but keep hierarchical subgraph restrictions, the model becomes strictly weaker than HE-GNN.
> > >
> > > To see this, consider the graphs $G_1, G_2$ where $G_1$ consists of an 8-cycle plus one more node connected to all these 8 nodes; and $G_2$ consists of two 4-cycles plus one more node connected to all these 8 nodes.
> > >
> > > Color-refinement (or standard GNNs) cannot distinguish the two graphs. Further, every radius $r$ subgraph of $G_i$ with $r\geq 2$ is the full graph $G_i$, while the radius $1$ subgraphs of $G_1$ are isomorphic to the corresponding radius $1$ subgraphs of $G_2$. It follows that a variant of HES-GNN with nested message-passing/refinement but without individualization cannot distinguish the graphs.
> > >
> > > However, the nodes that lie on the $4$-cycles of $G_2$ satisfies $\downarrow_x (\Diamond \Diamond(\neg x \wedge \Diamond^{\geq3} \Diamond x))$, while no node in $G_1$ satisfies this formula, showing that HE-GNN with depth $1$ separates the graphs. We would not know how to prove this separation easily without using the logical characterization.
> > >
> > > 4. **Logic  vs Learning**
> > >
> > > We believe trainability of even standard GNNs to be a relevant but understudied topic. Our expressiveness results do not have implications for trainability, which is why we have included practical experiments.

---

> > > > ### Comment · Reviewer_Ru4p · 2025-08-06
> > > > **Thanks for Authors**
> > > >
> > > > I appreciate the authors' thorough and well-reasoned responses. Most of my theoretical concerns were addressed with clarity and rigor. Thus, I will raise my score accordingly.

---

### Official Review · Reviewer_ZB9j · 2025-06-30

**Clarity:** 3
**Significance:** 2
**Originality:** 2
**Rating:** 4
**Confidence:** 4

**Summary:**

In this paper, the authors propose a graph neural network (GNN) architecture HE-GNN that goes beyond the expressivity limitations (i.e. 1-WL) of vanilla GNNs. This is then extended by allowing the model to consider only ego-nets of radius r, which improves the computational complexity. In particular, the proposed methods exploit the paradigm of Individualization-Refinement used by many subgraph isomorphism solvers. Most of the theoretical results on the graph separating power of the proposed architecture are obtained via logical characterizations, and the results are compared to some of the existing literature on higher-order GNNs and subgraph GNNs. Finally, some proof-of-concept experimental results are provided.

**Questions:**

- the authors should argue why a logic-based approach was chosen to describe the expressivity of the proposed architecture.
- can you discuss the memory and time complexity of the proposed method?

---
Finally, I wish to remark that if the authors address my concerns (missing related work, more justification for logic-based approach, and more thorough experimental evaluation), I would be willing to increase my score.

**Ethical Concerns:**

["NO or VERY MINOR ethics concerns only"]

**Final Justification:**

Overall, I am quite convinced by the rebuttal. The authors agreed to fix some clarity issues, improve the lacking literature review and perform some additional experiments (although they have not yet reported the results). I will raise my score from 3 to 4.

**Limitations:**

Yes.

**Paper Formatting Concerns:**

None.

**Quality:**

3

**Strengths And Weaknesses:**

Strengths:
- the theoretical results seem to be correct. Moreover, the use of logic, while it has already been used in some works on GNNs, is an interesting approach, and the graph learning community can benefit from it.
- The introduction of bounded radius ego-nets is an interesting and effective approach to reduce the model complexity.
- The comparison with higher-order and subgraph GNNs is well written and provides interesting insights.
- Code is provided and will be made public.

Weaknesses:
- The literature review is largely insufficient, as the work is not situated within the quite rich literature of individualization-refinement approaches in graph learning. I name a few here:
    - [1] provides a very general framework (IRNI) for endowing GNNs with individualization-refinement features, allowing to obtain full graph separating power. This work (and [2]) should be discussed thoroughly, as it is the closest to what the present work proposes.
    - [2] provides a learning-theoretic analysis of methods that individualize nodes as a way of improving expressivity, and proposes the Tinhofer encoding (based on the individualization-refinement framework) as a compromise between expressivity and generalization. Moreover, the work proposed a bounded-radius ego-net architecture EGO-NN. What are the differences with HES-GNNs?
    - [Haroon Dupty and Lee] (cited but discussed too briefly) and [3] provide methods to integrate the individualization refinement framework into GNN architectures.
    - [4] studies the expressive power of GNNs endowed with individualization methods from the prespective of logic, like the present work. The differences in the two approaches should be highlighted.
    - Several other works propose methods to individualize nodes to improve expressivity. This technique was first introduced in [5], and for other works check related work sections of [1] and [2].
- The use of logic to prove graph separation results, while interesting, is not properly justified. I suspect that easier proofs with tools that are more accessible to the machine learning and graph learning community could lead to the same results. Therefore, the authors should argue why a logic-based approach is to be preferred in this case.
- The high memory costs (quadratic) and time complexity (exponential in the depth d) make this architecture, while inteesting from the theory perspective, unlikely to be adopted in practice. Moreover, these time and space complexities should be discussed directly in the methods sections, rather than only in the limitations.
- The experimental validation is quite unsatisfactory.
    - The experimental framework is not properly defined neither in the main paper nor in the appendix. For example: does the test set include the same isomorphism classes of the training set in the strongly regular experiment? If so, why is there is a gap between train and test performance?
    - On the strongly regular dataset, it would be interesting to compare HES-GNNs to other individualization-refinement methods such as the Tinhofer encoding [2] or IRNI [1].
    - Only one real-world dataset (ZINC) is used, and the proposed baselines are quite weak. Comparing to the aforementioned methods or to node-marking-based subgraph GNNs would be interesting.
- The definition of HE-GNNs and HES-GNNs by bullet points is somewhat imprecise. Saying that a HE-GNN of nesting depth $d>0$ is a tuple $(B, C)$ with $B$ a HE-GNN of nesting depth $d-1$ and $C$ a GNN is not really enough. If the model just ignored $B$, we would get a normal GNN. I think run_A should be included in the definition (something like “a HE GNN of nesting depth $d>0$ is a tuple $(B,C)$ […] such that embeddings are updated according to run_A“)

---

- [1] Franks et al. A Systematic Approach to Universal Random Features in Graph Neural Networks. TMLR 2023.
- [2] Pellizzoni et al. On the Expressivity and Sample Complexity of Node-Individualized Graph Neural Networks. NeurIPS 2024.
- [3] Guo et al. Improving the Expressive Power of Graph Neural Network with Tinhofer Algorithm. 2021. (preprint)
- [4] Abboud et al. The surprising power of graph neural networks with random node initialization. IJCAI 2021.
- [5] Sato et al. Random features strengthen graph neural networks. SDM 2021.

---

> ### Author Rebuttal · Authors · 2025-07-30
>
> We thank the reviewer for their constructive feedback, and have revised our manuscript to address the points raised. We appreciate that the reviewer considers our logical approach and comparison with existing architectures insightful and beneficial to the graph learning community.
>
> **Literature review**
>
> The reviewer points to several relevant references that warrant a detailed comparison. We have added a *Related work* section, making our contribution in relation to these references precise. Below, we give an overview of the distinctions between our work and [1–4], as requested by the reviewer.
>
> Crucially, these works all cover probabilistic architectures, trading full isomorphism equivariance for equivariance in expectation in order to gain tractability and expressivity. As a result, models presented in [1–4] are less suited for comparison with isomorphism-equivariant classifiers, such as those defined by the WL hierarchy or logics.
>
> The results in these works hence cover different questions than ours. Franks et al. [1] investigate function approximation, while Pellizzoni et al. [2] analyze bounds on VC-dimension. Guo et al. [3] provide experimental validation but no theoretical results. [1], [2], and [3] all use node individualization, but do so by selecting a single path in the Individualization-Refinement tree using a probabilistic method such as Tinhofer. None of these approaches use a recursive message-passing procedure similar to HE-GNNs, which aggregate over all leaves of an IR-tree in a fully equivariant manner.
>
> **Use of logic**
>
> We find that logical characterizations provide a formal and explainable perspective on graph learning, which has yielded several results in recent years.
>
> One example of this is given in reference [4] listed by the reviewer. Here, Abboud et al. prove the universal separating power of GNNs with random node initialization using a logical characterization by Barcelo et al. [6]. Additional examples are given by Tena Cucala et al. [7], who use conjunctive query characterizations of monotonic GNNs to generate automatic explanations, and Adam-Day et al. [8], who obtain convergence laws for GNNs via term languages.
>
> In addition, logical characterizations can allow for intuitive proofs. For example, in Theorem 5.11 we distinguish HES-GNN from $k$-GNN by showing that the former can recognize colorful distance-2 cliques. Reasoning in terms of nested message-passing aggregations would be tedious here. Instead, we give a single intuitive GML$(\downarrow_W)$ formula, of which one can directly verify that it recognizes such a clique.
>
> We take this comment seriously, as it may be shared by more researchers in the graph learning community, and have added further justification for our logical approach in the introduction.
>
> - [6] Barceló, P., Kostylev, E. V., Monet, M., Pérez, J., Reutter, J., & Silva, J. P. (2020, April). *The logical expressiveness of graph neural networks.* In 8th International Conference on Learning Representations (ICLR 2020).
> - [7] Tena Cucala, D. J., Cuenca Grau, B., Kostylev, E. V., & Motik, B. (2022). *Explainable GNN-based models over knowledge graphs.*
> - [8] Adam-Day, S., Benedikt, M., Ceylan, I., & Finkelshtein, B. (2024). *Almost surely asymptotically constant graph neural networks.* Advances in Neural Information Processing Systems, 37, 124843–124886.
>
> **Experimental validation**
>
> In the SRG experiment, train and test sets indeed include the same isomorphism classes. The reviewer has identified an issue due to an unexpected interaction with PyTorch `.eval()`. The issue has been resolved. Train and test performance now match, and the manuscript explains the experimental setup more clearly. Our conclusion from this experiment is not affected.
>
> We have focused on formalizing and presenting the capacity of HE-GNN to express existing architectures, and recognize that further experimental validation would strengthen the paper. We will expand the experimental section as follows:
>
> 1. We will compare performance with more existing architectures. We note that [1] does not provide code for comparison, but comparison with Tinhofer is feasible. We will do our best to compare performance on ZINC with more node-marking based subgraph GNNs.
>
> 2. We will provide more results with varying $d, r$, to show how HE-GNNs transition from standard message-passing networks to more expressive architectures in terms of both accuracy and running time.
>
> **Final remarks**
>
> We have made the definition of HE-GNN more explicit. Further, we have made the complexity analysis of HE-GNN more precise, and moved it from the comparison with higher-order GNN to the section that defines the model. We note that the crucial distinction with higher-order GNNs in terms of complexity is made in section 5.3. HE-GNNs with depth $d$ store at most $|G|^2$ features, but they perform $O(|G|^d)$ rounds of message-passing.

---

> > ### Comment · Reviewer_ZB9j · 2025-08-04
> >
> > Thank you for your response.
> >
> > > Literature review
> >
> > That's great. Be sure to include these references, as well as the other ones suggested by the other reviewers. as pointed out by Reviewer bFLP, the literature review was quite poor.
> > Also, note that on WL-amenable graphs (most of real-world graphs), IR procedures are permutation equivariant, so your statement on "probabilistic architectures" is not completely correct.
> >
> > > Use of logic
> >
> > Resolved. Please include this discussion, i don't think the graph learning community is very familiar with logic.
> >
> > > Experimental validation
> >
> > Do you already have some preliminary results? (I understand that one week is not a lot of time to get results.)
> >
> > > Final remarks
> >
> > That's great.
> >
> > Overall, I am quite convinced by the rebuttal. I will raise my score.

---

> > > ### Author Response · Authors · 2025-08-04
> > >
> > > Thank you for the addition. We are aware of this invariance property and will make a correct statement in the paper.
> > >
> > > We are indeed still setting up the additional experiments. We have found that HE-GNN achieves 100% accuracy on the SRG experiment (on both train and test set), but have not yet compared this with Tinhofer.

---

> > > ### Author Response · Authors · 2025-08-08
> > >
> > > We have preliminary results from one experiment: Tinhofer does not generate canonical orderings on the strongly regular graphs we considered in the paper. Hence, GNNs augmented with Tinhofer ordering reach 100% accuracy on the training set, but only 33% accuracy on the test set.
> > >
> > > For these graphs, HES-GNN with $r \geq 2, d=2$ reaches 100% accuracy on both training and test set.

---

### Official Review · Reviewer_bFLP · 2025-07-01

**Clarity:** 4
**Significance:** 2
**Originality:** 2
**Rating:** 4
**Confidence:** 5

**Summary:**

This paper introduces Hierarchical Ego GNNs (HE-GNNs), which augment standard message-passing GNNs by recursively “individualiing” a chosen node, marking it uniquely, and re-running a base GNN to capture increasingly fine-grained local structure. By nesting this individuation process to depth $d$, HE-GNNs form a strict hierarchy: at depth $d$, they match the distinguishing power of $d$-nested graded hybrid logic with variable binders, and as $d\to\infty$, they can separate any pair of non-isomorphic graphs. The authors provide rigorous logical characterizations showing that depth-1 HE-GNNs subsume common subgraph-based GNNs, that $d$ rounds of HE-GNN individuation lower-bound $d$ steps of the Individualization-Refinement paradigm, and that with sufficient nesting they can simulate homomorphism-count-augmented GNNs. An illustrative experiment on ZINC-12k and graph isomorphism testing capabilities concludes the paper.

**Questions:**

1. How does the recursive individualization in HE-GNNs compare to the Individualization–Refinement framework used in “A Systematic Approach to Random Data Augmentation on Graph”?  https://arxiv.org/abs/2112.04314
2. Are there known expressiveness or complexity results for graded hybrid modal logics with variable binders that you can leverage? Grounding HE-GNN’s logical characterization in a well-studied fragment would broaden its impact.

**Ethical Concerns:**

["NO or VERY MINOR ethics concerns only"]

**Final Justification:**

In its favour, the paper stands out in clarity and formal rigor, and it offers an elegant use of logic to model hierarchical message passing, which is then grounded in a corresponding neural architecture. However, this logical framework yields fewer theoretical (and practical) insights than prior hybrid‐logic approaches, since little of the hybrid logic world seems transferable to the neural setting. Coupled with a rather weak experimental evaluation (the work’s primarily theoretical focus) these factors lead me to recommend a weak accept.

**Limitations:**

Yes

**Paper Formatting Concerns:**

No issues.

**Quality:**

3

**Strengths And Weaknesses:**

**Strengths**
1. The paper is very well written, with precise definitions and a coherent, logical progression.
2. Prior approache are surveyed thoroughly and compared clearly.
3. Adopting graded hybrid modal logic's variable binders to model node individualization is elegant and conceptually well motivated.
4. The paper delivers an extensive number of expressiveness results—characterizing HE-GNNs by depth $d$ and radius $r$ and situating them relative to competing paradigms.

**Weaknesses**
1. The modal-logic characterization focuses on graph distinguishability; proving uniform expressiveness (i.e., the ability to compute arbitrary logical queries) would significantly strengthen the contribution.
2. Theorem 5.10 (relating HE-GNNs to higher-order GNNs) may also be derived via the Geerts & Reutter ICLR 2022 analysis. It remains unclear whether the logic perspective offers insights beyond serving as an alternative proof technique.
3. While Section 5.4 sketches the relationship, a tighter, perhaps quantitative, comparison to state-of-the-art subgraph-augmented GNNs would be valuable.
4. The empirical evaluation, though illustrative, remains limited. More diverse benchmarks or ablations (e.g., varying $d$, $r$) would bolster confidence in practical gains.

---

> ### Author Rebuttal · Authors · 2025-07-30
>
> We thank the reviewer for their feedback. We have considered all comments and revised our manuscript accordingly. We note that the reviewer values the extensive theoretical comparison of HE-GNN with existing paradigms and the introduction of hybrid modal logic characterizations. Indeed, we intend this work as opening up questions in graph learning to the logic community.
>
> **Uniform characterizations**
>
> Addressing the first raised weakness, we agree with the reviewer that uniform translations are essential in understanding expressive power. Our work does in fact provide uniform translations from GML$(\downarrow)$ to HE-GNN and from GML$(\downarrow_W)$ to HES-GNN, showing that the graph learning architectures can compute arbitrary logical queries in their characterizing languages. This is explained below theorems 3.3 and 4.1.
>
> In the converse direction, we provide a uniform translation only over bounded degree graphs. General uniform translations from the learning architectures to the logics as we defined them do not exist. For example, a GNN can compute whether a node has an odd number of neighbors, which is not expressible in first-order logic. It would be interesting to develop a logic that can (uniformly) represent HE-GNN classifiers by allowing richer forms of arithmetic, following Grohe [1] and Benedikt et al. [2], or to obtain uniform characterizations relative to first-order logic.
>
> [1] Grohe, M. (2024). The descriptive complexity of graph neural networks. TheoretiCS, 3.
>
> [2] Benedikt, M., Lu, C. H., & Tan, T. (2024). Decidability of graph neural networks via logical characterizations. arXiv preprint arXiv:2404.18151.
>
> **Alternative approach for 5.10**
>
> In the second raised weakness the reviewer remarks that 5.10 might also be derived via the Tensor-Language analysis from Geerts & Reutter. Writing a HE-GNNd$(d)$ update as a TL-expression with $d+1$ variables would yield the same subsumption of separating power by that of $d$-GNN.
>
> We expect that a proof in this direction is possible, but not trivial. A TL expression describing a single HE-GNNd$(d)$ update must include message-passing sequences on $|G|^d$ distinct graphs, aggregated hierarchically. Conversely, with the GML$(\downarrow)$ characterization in hand the standard translation to first-order logic directly yields the desired result, benefiting from the longstanding relations between modal and first-order logic.
>
> We note that logical characterizations have yielded significant insights in graph learning. For example, Abboud et al. [3] proved the universal separating power of GNNs with random initialization using a logical characterization by Barceló et al. [4], Tena Cucala et al. [5] used conjunctive query characterizations of monotonic GNNs to generate automatic explanations for predictions, and Adam-Day et al. [6] obtained various convergence laws for GNNs via term languages.
>
> [3] Abboud et al. *The surprising power of graph neural networks with random node initialization.* IJCAI 2021.
>
> [4] Barceló, P., Kostylev, E. V., Monet, M., Pérez, J., Reutter, J., & Silva, J. P. (2020, April). *The logical expressiveness of graph neural networks.* In 8th International Conference on Learning Representations (ICLR 2020).
>
> [5] Tena Cucala, D. J., Cuenca Grau, B., Kostylev, E. V., & Motik, B. (2022). *Explainable GNN-based models over knowledge graphs.*
>
> [6] Adam-Day, S., Benedikt, M., Ceylan, I., & Finkelshtein, B. (2024). *Almost surely asymptotically constant graph neural networks.* Advances in Neural Information Processing Systems, 37, 124843–124886.
>
> **Empirical validation**
>
> We have presented HE-GNN primarily as a foundational model, unifying existing architectures in terms of separating power. We recognize that further experimental validation would strengthen the paper and aim to expand the experimental section as follows:
>
> 1. We will compare performance on ZINC with other node-marking based subgraph GNNs, addressing weakness 3 raised by the reviewer.
>
> 2. We will provide more results with varying $d,r$, to show how HES-GNNs transition from standard message-passing networks to more expressive architectures in terms of runtime and accuracy, in response to weakness 4 raised by the reviewer.
>
> **Response to questions**
>
> **Relation to Franks et al., 2022**
>
> The reviewer asks how our work differs from that of Franks et al., who also study GNNs with node encodings derived from individualization. Their approach uses random walks to probabilistically sample a path in an Individualization-Refinement tree, trading full isomorphism equivariance for equivariance in expectation in order to gain tractability. In contrast, our method applies recursive message-passing to aggregate over all leaves of the IR-tree in a fully equivariant manner.
>
> The approach by Franks et al. is hence less suited for comparison with equivariant classifiers, such as those defined by the WL hierarchy or logics. The authors accordingly investigate function approximation rather than separating power.
>
> We now make this distinction clear in a new *Related work* section.
>
> **Applicable results in graded hybrid modal logic**
>
> The reviewer asks if graded hybrid modal logic is a well-studied fragment, so that the characterization of HE-GNN in this logic yields additional results.
>
> The decidability and complexity of reasoning tasks have been studied in detail for hybrid logics. Although the satisfiability problem for hybrid logics is undecidable, it can be made decidable by either forbidding certain specific syntactic nesting patterns, or through semantic restrictions. See ten Cate & Franceschet [7]. While these results apply to hybrid formulas without graded modalities, we believe the analyses can be extended to the case with graded modalities as well. Therefore, as suggested by the reviewer our logical characterizations may indeed generate additional insight into sources of complexity when it comes to reasoning about GNNs.
>
> We have added this viewpoint to the introduction as further justification for our approach.
>
> [7] Ten Cate, B., & Franceschet, M. (2005, August). *On the complexity of hybrid logics with binders.* In International Workshop on Computer Science Logic (pp. 339–354). Berlin, Heidelberg: Springer Berlin Heidelberg.

---

> > ### Comment · Reviewer_bFLP · 2025-08-02
> > **Response to Rebuttal**
> >
> > Thanks for your response.
> > - Regarding uniform expressiveness, I was indeed referring to the general case rather than the specific translation results presented in the paper.
> > - On TL: intuitively, a parameterised expression with $d$ extra variables should be able to perform message passing over all $d$-individualised graphs. In practice, working directly with higher-order GNNs, or the corresponding TL fragmen, can be more transparent than encoding everything in pure logic.
> > - As the author acknowledge, the experimental evaluation needs expansion and deeper analysis.
> > - Thank you for clarifying the connection to Franks’ work; please be sure to include all relevant citations in the paper (also those mentioned by the other reviewers, you indeed are missing crucial references).
> > - Finally, on graded hybrid modal logic, I’m still unclear whether your perspective yields new theoretical insights. In contrast, the seminal GNN paper’s link to the well-studied $C_2$ logic immediately provided results for graph learning.

---

> > > ### Author Response · Authors · 2025-08-04
> > >
> > > Thank you for the addition.
> > >
> > > On TL, to capture HEGNN an expression should indeed perform message-passing over all $d$-individualized graphs, but the output should further be aggregated hierarchically. I.e., on each depth $d-1$ graph, the output from $|G|$ further individualizations are used in message-passing. The output features from this depth are then aggregated on the depth $d-2$ graphs, etc.
> > >
> > > In this case, the standard translation from modal logic is applicable off-the-shelf. We understand that a proof via TL would be more satisfying for some readers.
> > >
> > > We note that the logical characterization has yielded several theoretical results in the paper, such as situating HEGNN in the WL-hierarchy, and is a useful tool for analysis in general. One additional example is that it allows us to solve a question posed by reviewer Ru4p in the follow-up.
> > >
> > > We have already provided some references where similar characterizations have proven useful, but are not aware of a direct result on, e.g., decidability of this fragment, besides what we have written in the rebuttal.

---

> > > > ### Comment · Reviewer_bFLP · 2025-08-05
> > > >
> > > > Thanks for the clarification, the hierarchical message passing may indeed make this translation unnecessarily complicated.

---

### Official Review · Reviewer_et5K · 2025-07-02

**Clarity:** 3
**Significance:** 3
**Originality:** 3
**Rating:** 4
**Confidence:** 4

**Summary:**

The authors study the expressivity of graph neural networks (GNNs) with hierarchical node individualisation. For that they propose a new model of GNNs that they name Hierarchical Ego Graph Neural Networks (HE-GNNs) and show how its expressivity can be captured by graded hybrid modal logic. The expressivity characterisation follows the line of work initiated by Barcelo et at. [4] where the expressivity of GNNs was related to graded modal logic. Hence the main contribution of the authors here is to extend this connection to GNNs with node initialisation, which is in the logical side represented by the binder operator of hybrid logic. They also study a variant of HE-GNNs, where message passing related to node initialisations is restricted to some k-hop neighbourhood of the initialised node and obtain a related logical characterisation to that model, and an expressivity hierarchy related to k and initialisation depth d.

In addition the authors introduce a variant of the 1-WL algorithm with node initialisation (WL-IR) and show that the separating power of depth d HE-GNNs is limited below by the separating power of WL-IR with d node initialisations (WL-IR-d). Note that, over graphs of size at most d, WL-IR-d is able to separate all non-isomorphic graphs.

They empirically confirm that depth 2 HE-GNNs have improved performance to distinguish strongly regular graphs in comparison to common subgraph architectures.

**Questions:**

1. What is the benefit or difference of your HE-GNNs of depth d compared to a simpler variant that would simply run a GNN on the |G|^d many copies of the graph with all possible and then have some aggregation via a GNN. That is, what is the benefit of the d-layer recursive definition.

2. Could you say something on the topic of training  HE-GNNs of depth d. Can this be done in practise?

**Ethical Concerns:**

["NO or VERY MINOR ethics concerns only"]

**Final Justification:**

This is a solid well written paper that advances our understanding in the expressivity of GNNs and contributes to the growing literature connecting logics and neural networks. The results are not ground breaking, and as mentioned by others, the experimental part is quite limited. Hence, I think the rating of borderline accept is well justified.

**Limitations:**

Yes.

**Paper Formatting Concerns:**

None.

**Quality:**

3

**Strengths And Weaknesses:**

Strengths:

- The topic of the paper is timely and the results are interesting. The paper is very well written. I did not find any major technical issues, though I did not check all the details from the appendix.


Weaknesses:
- The presentation of HE-GNNs is quite compact. A worked out example of a simple HE-GNNs  would have been nice to have.
- There is not much discussion whether their GNN model is useful in practice.
- In isolation the results are not that surprising, but the authors have done a comprehensive job in developing matching models of GNNs, logic, and WL-algorithm.


Minor comments:

- l. 151. Typo: "in in"
- l. 161. Technically, in GML formulas with free nominals, you cannot eliminate the "@" operator.
- l. 186. Typo: Period missing after the syntax of the logic.

---

> ### Author Rebuttal · Authors · 2025-07-30
>
> We thank the reviewer for their constructive feedback, and have revised our manuscript accordingly. We appreciate that the reviewer considers the paper to be well-written, interesting, and sound.
>
> **Response to weaknesses**
>
> - We will add a simple example of a HE-GNN.
>
> - We have focused on formalizing and presenting the capacity of HE-GNN to express existing architectures, but this expressive power also leads to performance gains in practice. We will convey this more clearly by firstly adding a more precise complexity analysis, and secondly by extending the *Experiments* section. Specifically, we will attempt to compare performance on ZINC with other modern subgraph GNNs.
>
> - We thank the reviewer for spotting the typos, which have been resolved.
> The reviewer correctly notes that @ cannot be eliminated in formulas with free variables. The paper only considers sentences (line 148), and we have clarified the claim on line 161 accordingly.
>
> **Response to questions**
>
> **Question 1**
>
> The first question asks for the distinction between HE-GNN and running a GNN on the $|G|^d$ many copies of the graph with all possible colorings, and aggregating via a GNN.
>
> This is an interesting remark that ties into the discussion at the end of Section 5.4, since the model suggested by the reviewer can be implemented as an OSAN. It is not directly clear how the two models differ in terms of separating power, which also depends on the exact method used to aggregate over all colorings. We appreciate the question, which deserves further investigation.
>
> A crucial difference is that combining features derived from $|G|^d$ colorings with a GNN requires exponential space complexity in $d$, while our hierarchical message-passing scheme requires only a constant number of stored features in $d$.
>
> **Question 2**
>
> The second question asks whether HE-GNNs can be trained in practice.
>
> This is partly answered by the experimental section. We train HES-GNNs with $d \leq 2$, going beyond the separating power of $3$-GNNs on a single GPU. For larger $d$, the runtime behavior requires restricting the radius $r$.
>
> We will further address this question quantitatively in the paper by adding an ablation study, comparing the runtime and performance of HES-GNN for varying $d$ and $r$.

---

> > ### Comment · Reviewer_et5K · 2025-08-03
> > **Response for rebuttal**
> >
> > Thank you for your rebuttal. I will retain my score for now.

---

### Official Review · Reviewer_Gw3G · 2025-07-08

**Clarity:** 3
**Significance:** 2
**Originality:** 2
**Rating:** 4
**Confidence:** 4

**Summary:**

This paper introduces Hierarchical Ego Graph Neural Networks (HE-GNNs), a class of GNNs inspired by the individualization-refinement (IR) paradigm used in graph isomorphism testing. The authors provide a logical characterization of HE-GNNs (and a subgraph-restricted variant), relate their expressive power to higher-order GNNs, subgraph-GNNs, and GNNs with homomorphism counts, and demonstrate empirically that shallow HE-GNNs offer improved separating power over traditional GNNs. The authors conducted experiments on ZINC-12k and synthetic regular graphs.

**Questions:**

- Could you please conduct experiments on a broader set of real-world datasets and compare HE-GNNs with additional, recently proposed expressive GNN baselines? I leave the choice of datasets and baselines to the authors’ discretion, but a more comprehensive empirical evaluation would greatly strengthen the paper.

- The reference list is very weak. Please conduct a thorough literature review. Some related works can be found below:

Huang, Yinan, William Lu, Joshua Robinson, Yu Yang, Muhan Zhang, Stefanie Jegelka, and Pan Li. "On the stability of expressive positional encodings for graphs." arXiv preprint arXiv:2310.02579 (2023).

Kanatsoulis, Charilaos I., Evelyn Choi, Stephanie Jegelka, Jure Leskovec, and Alejandro Ribeiro. "Learning Efficient Positional Encodings with Graph Neural Networks." arXiv preprint arXiv:2502.01122 (2025).

Black, Mitchell, Zhengchao Wan, Gal Mishne, Amir Nayyeri, and Yusu Wang. "Comparing graph transformers via positional encodings." arXiv preprint arXiv:2402.14202 (2024).

Kanatsoulis, Charilaos, and Alejandro Ribeiro. "Counting graph substructures with graph neural networks." In The twelfth international conference on learning representations. 2024.

Lim, Derek, Joshua Robinson, Lingxiao Zhao, Tess Smidt, Suvrit Sra, Haggai Maron, and Stefanie Jegelka. "Sign and basis invariant networks for spectral graph representation learning." arXiv preprint arXiv:2202.13013 (2022).

Nerem, Robert R., Samantha Chen, Sanjoy Dasgupta, and Yusu Wang. "Graph neural networks extrapolate out-of-distribution for shortest paths." arXiv preprint arXiv:2503.19173 (2025).

**Ethical Concerns:**

["NO or VERY MINOR ethics concerns only"]

**Final Justification:**

The authors have addressed most of my concerns and have committed to expanding their experimental evaluation. Assuming these improvements are incorporated into the final version, I believe the paper will be sufficiently strengthened and is worthy of publication as a borderline paper.

**Limitations:**

Please see previous sections

**Quality:**

2

**Strengths And Weaknesses:**

**Strengths**

- Theoretical framework is rigorous and unifies connections to higher-order GNNs, subgraph-GNNs, and homomorphism-count GNNs.
- Logical analysis is novel and deepens understanding of GNN expressiveness.

**Weaknesses**

- Experimental validation is quite limited, focusing only on ZINC-12k and small synthetic datasets. There is no evaluation on larger or more diverse real-world benchmarks, nor ablations on computational overhead or scalability.
- There are limited comparisons with related literature and several omitted works. Please see next section for more details

---

> ### Author Rebuttal · Authors · 2025-07-30
>
> We thank the reviewer for their constructive feedback and have addressed all comments in our revised manuscript. We appreciate the reviewer's assessment that our theoretical framework deepens the understanding of GNN expressiveness and unifies several existing architectures, which we indeed consider to be the core contribution of this work.
>
> **Experimental validation**
>
> We have presented HE-GNN primarily as a foundational model, and recognize that further experimental validation would strengthen the paper. Addressing the first weakness and question raised by the reviewer, we aim to expand the experimental section as follows:
>
> 1. We will compare performance with more existing architectures. Specifically, we aim to complement the SRG experiments with another Individualization-Refinement technique such as Tinhofer, and to compare performance on ZINC with other modern node-marking-based subgraph GNNs.
>
> 2. We will add an ablation study varying $d$ and $r$, illustrating how HES-GNN transitions from standard message-passing networks to more expressive architectures in terms of both accuracy and running time.
>
> **Literature review**
>
> In response to the second weakness and question raised by the reviewer, we have added a *Related work* section, covering part of the literature listed by the reviewer, as well as suggestions from other reviewers. Four of the references cited by the reviewer use positional encodings derived from the Laplacian of the input. This approach differs from the line of work leading up to our paper and addresses slightly different questions, such as perturbation stability. The work by Black et al. is closest to ours, since it analyzes separating power via extensions of the WL algorithm, and we agree that exploring further connections between these two approaches is valuable and interesting.

---

> > ### Comment · Reviewer_Gw3G · 2025-08-05
> >
> > Thank you for your response. Could you please share the updated experiments and the revised parts of the manuscript? The current reply suggests some changes, but without seeing them, I am unable to properly evaluate the changes.

---

> > > ### Author Response · Authors · 2025-08-08
> > >
> > > We understand your concern, but the rules and instructions we were given forbid uploading revisions of the manuscript in the discussion stage. We give an outline of currently implemented changes related to your comments.
> > >
> > > **Literature**
> > >
> > > In the updated manuscript, we reference GNNs enriched with positional encoding:
> > >
> > > - Dwivedi, V. P., Luu, A. T., Laurent, T., Bengio, Y., & Bresson, X. (2021). Graph neural networks with learnable structural and positional representations. arXiv preprint arXiv:2110.07875.
> > >
> > > - Rampášek, L., Galkin, M., Dwivedi, V. P., Luu, A. T., Wolf, G., & Beaini, D. (2022). Recipe for a general, powerful, scalable graph transformer. Advances in Neural Information Processing Systems, 35, 14501-14515.
> > >
> > > - Huang, Yinan, William Lu, Joshua Robinson, Yu Yang, Muhan Zhang, Stefanie Jegelka, and Pan Li. On the stability of expressive positional encodings for graphs.
> > >
> > > - Kanatsoulis, Charilaos I., Evelyn Choi, Stephanie Jegelka, Jure Leskovec, and Alejandro Ribeiro. Learning Efficient Positional Encodings with Graph Neural Networks.
> > >
> > > GNNs with random features:
> > >
> > > - Abboud et al. The surprising power of graph neural networks with random node initialization. IJCAI 2021.
> > >
> > > - Sato et al. Random features strengthen graph neural networks. SDM 2021.
> > >
> > > And GNNs with features obtained by probabilistic individualization:
> > >
> > > - Franks et al. A Systematic Approach to Universal Random Features in Graph Neural Networks. TMLR 2023.
> > >
> > > - Pellizzoni et al. On the Expressivity and Sample Complexity of Node-Individualized Graph Neural Networks. NeurIPS 2024.
> > >
> > > - Guo et al. Improving the Expressive Power of Graph Neural Network with Tinhofer Algorithm. 2021. (preprint)
> > >
> > > **Experiments**
> > >
> > > We have currently completed one experiment: comparing HES-GNN with Tinhofer individualization-refinement, as suggested by reviewer ZB9j. HES-GNN with $d=2, r \geq 2$ distinguishes the strongly regular graphs with 100% accuracy in both training and test set.
> > >
> > > Tinhofer fully individualizes the graphs, but it does not produce a canonical isomorphism-invariant ordering. GNNs with Tinhofer hence reach 100% accuracy on the training set, but only 33% on the test set.
> > >
> > > We are still setting up further comparisons on ZINC and ablation studies.

---

### Note · Authors · 2025-08-12

We have no further additions.

We sincerely appreciate all reviewers' efforts and interest, and the insightful discussion that has significantly enhanced the relevance of this work.

---

### Decision · Program_Chairs · 2025-09-17

**Decision:**

Accept (poster)

**Comment:**

The paper examines the expressive power of Graph Neural Networks (GNNs) utilizing node individualization. It introduces a new model, analyzes its logical expressive power, and establishes interesting connections to previous expressive models, such as subgraph GNNs. Several issues were raised during the discussion, all of which were resolved satisfactorily. In the reviewer–area chair discussion, a consensus was reached that the paper is a valuable addition to the conference. I am pleased to recommend its acceptance and encourage the authors to include all the materials requested by the reviewers.